# Precision screening identifies mitoxantrone as a multitarget inhibitor in ageing-associated cancers with extensive computational validation and doxorubicin comparison

Mohammed H. Al-Qahtani[ORCID][1,2]*, Mourad Assidi[1,2], Abdelbaset Buhmeida[1,2], Asma Almuhammadi[3], Peter Natesan Pushparaj[1,2], Nofe Ateq Alganmi[1,4]

1 Institute of Genomic Medicine Sciences, King Abdulaziz University, Jeddah, Saudi Arabia, 2 Department of Medical Lab Technology, Faculty of Applied Medical Sciences, King Abdulaziz University, Jeddah, Saudi Arabia, 3 Department of Biological Sciences, Faculty of Science, King Abdulaziz University, Jeddah, Saudi Arabia, 4 Computer Science Department, Faculty of Computing and Information Technology, King Abdulaziz University, Jeddah, Saudi Arabia

* mhalqahtani@kau.edu.sa, qahtani_i@hotmail.com

## Abstract

Ageing-associated cancers are characterised by the dual hallmarks of persistent DNA damage and the ability of tumour cells to escape senescence checkpoints, which drive genomic instability and uncontrolled proliferation. In this study, we identified crucial proteins with PDB IDs—2YEX (Chk1 kinase), 4HG7 (MDM2 E3 ubiquitin ligase), 4JSX (mTOR kinase domain), and 5DS3 (PARP-1 DNA repair enzyme)—involved in ageing-related cancers and performed docking studies with Extra Precision (XP) followed by MM-GBSA-based pose processing against the FDA-approved DrugBank library (LigPrep: 10907 compounds). The extensive docking computations identified many good candidates; however, Mitoxantrone emerges as the topmost candidate with docking scores of −6.23 to −16.044 Kcal/mol and MM-GBSA score of −49.19 to −85.14 Kcal/mol, which currently is being used to treat advanced prostate cancer and acute nonlymphocytic leukaemia (ANLL) and would be easier to repurpose to other ageing-related cancers. Mitoxantrone also emerges as a better candidate compared to the control drug Doxorubicin. Further, the complex of all 4 proteins with Mitoxantrone was taken for interaction fingerprints and found that the most interacting residues with counts were 6GLY, 6VAL, 5GLU, 5LEU, 4ALA, 3ASP, and 3TYR, among others. The pharmacokinetics and Density Functional Theory computations further support Mitoxantrone as a potential candidate. We also performed a 5-nanoseconds (ns) WaterMap for hydration site identification and the role of water in stabilisation of the complex, followed by a 100 ns MD Simulation that resulted in stable deviation and fluctuations mostly under <2Å and a web of simulation interactions making the complex stable. Furthermore, the same trajectories were used for the Binding Free and Total Complex Energies computations, revealing that the complexes were

**Data availability statement:** The Data used in this study are public, and the data generated during this analysis are in the paper as figures, tables and also provided a supplementary files that contains multiple sheets (also cited in the text).

**Funding:** This research work was funded by Institutional Fund Projects under grant number IFPNC: 009-117-2020. The authors gratefully acknowledge technical and financial support from the Ministry of Education and King Abdulaziz University, Jeddah, Saudi Arabia. The funders had no role in study design, data collection and analysis, decision to publish, or preparation of the manuscript.

**Competing interests:** The authors have declared that no competing interests exist.

**Abbreviations:** ADME, Absorption, Distribution, Metabolism, and Excretion; ALIE, Average Local Ionisation Energy; ANLL, acute nonlymphocytic leukaemia; BBB, blood–brain barrier; Chk, Checkpoint kinase1; Chk1, checkpoint kinase 1; D3 term, Dispersion corrections; DDR, DNA damage response; DFT, Density Functional Theory; DIIS, Direct Inversion in the Iterative Subspace; ESP, electrostatic potential; FDA, Food and Drug Administration; HOMO, Highest Occupied Molecular Orbital; HTVS, High-Throughput Virtual Screening; LUMO, Lowest Unoccupied Molecular Orbital; MD, molecular dynamics; MDM2, Mouse double minute 2 homolog; MEP, molecular electrostatic potential; MIFs, Molecular Interaction Fingerprints; MM-GBSA, Molecular Mechanics Generalised Born Surface Area; mTOR, Mechanistic Target of Rapamycin; MW, molecular weight; ns, nanoseconds; NTOs, Natural Transition Orbitals; PARP-1, Poly [ADP-ribose] polymerase 1; PDB, Protein Data Bank; PCM, Polarizable Continuum Model; PSA, polar surface area; RCSB, Research Collaboratory for Structural Bioinformatics; Rg, radius of gyration; RMSD, root mean square deviation; RMSF, root mean square fluctuation; SASP, senescence-associated secretory phenotype; SCF, Self-Consistent Field; SID, Simulation Interaction Diagram; SP, Standard Precision; TDDFT, Time-Dependent Density Functional Theory; TERT, telomerase reactivation; VSW, Virtual Screening Workflow; XP, Extra Precision; ΔG, free energy; ΔH, enthalpy.

stable. All the studies, from protein energies to docking to simulation and binding free energy, supported the stable complexes; however, experimental studies are necessary before their use.

---

## 1 Introduction

Cancer remains one of the most formidable challenges to human health, and its intricate association with ageing has become increasingly evident in recent decades [1,2]. As global life expectancy rises — for example, there are currently over 703 million people worldwide aged over 65, constituting 9.1% of the global population, with projections of 1.5 billion (≈ 15.9%) by 2050— the incidence and burden of cancer among older adults are escalating [3,4]. Indeed, in 2019, individuals aged 75 years and older accounted for approximately 28.6% of all new global cancer cases. Similarly, in 2022, an estimated 2.6 million new cancer cases (≈14% of total) and 2.1 million cancer deaths (≈22% of total) occurred in people aged 80 years or older [5,6]. Ageing is not simply the passage of chronological time, but a progressive decline in molecular integrity, cellular function, and systemic homeostasis [7]. This decline is driven by cumulative genomic damage, mitochondrial dysfunction, telomere attrition, epigenetic drift, and diminished repair capacity. Among the hallmarks of ageing, genomic instability and cellular senescence are particularly relevant to cancer development [8]. Genomic instability permits the accumulation of mutations, chromosomal rearrangements, and epigenetic changes that undermine cellular regulation. Senescence — a stable cell-cycle arrest responding to telomere shortening, oncogene activation, or oxidative/genotoxic stress — initially functions as a tumour-suppressive barrier by preventing propagation of damaged cells [9]. However, senescent cells often acquire a senescence-associated secretory phenotype (SASP), characterised by releasing pro-inflammatory cytokines, chemokines and growth factors that promote a tumour-permissive microenvironment in chronic settings [9,10]. Moreover, senescence escape via inactivation of tumour-suppressor pathways or telomerase reactivation can generate highly aberrant clones predisposed to malignant transformation [11].

Paradoxically, although impaired DNA repair accelerates oncogenic transformation, established cancer cells often become hyperdependent on specific DNA damage response (DDR) pathways for survival [12]. Due to relentless replication stress and oxidative insults, tumour cells upregulate DDR proteins such as ATM, ATR, and DNA-PKcs to sustain proliferation under genotoxic conditions [12]. Notably, cancers harbouring BRCA1 or BRCA2 mutations rely heavily on PARP1-mediated single-strand break repair to compensate for defective homologous recombination [12]. Pharmacological inhibition of PARP enzymes in such contexts induces synthetic lethality—a mechanism that selectively eradicates tumour cells unable to repair double-strand breaks. Similarly, checkpoint kinase 1 (Chk1) is a pivotal effector in the ATR-Chk1 signalling axis, orchestrating S-phase arrest and stabilising replication forks [9]. Tumours with elevated Chk1 expression display enhanced resistance to

DNA-damaging agents, underscoring its role as a survival determinant [13]. This adaptive hyperactivation of DDR components exemplifies the dualistic nature of repair mechanisms: while their dysfunction promotes oncogenesis, their sustained activation maintains tumour viability [9]. Consequently, targeting DDR nodes such as Chk1 and PARP1 has become a cornerstone of precision oncology. However, resistance frequently emerges due to compensatory pathway activation or mutational rewiring of DNA repair networks [14]. This limitation underscores the need for multitarget strategies simultaneously intercepting multiple survival axes within the DDR machinery [15,16].

In parallel, cellular senescence represents a crucial nexus linking ageing to cancer progression. Senescence is a stable, non-proliferative state triggered by diverse stimuli, including telomere erosion, oncogene activation, and oxidative or genotoxic stress [13]. Initially, senescence acts as a tumour-suppressive barrier by stopping the growth of damaged cells. This process is mainly controlled through the activation of tumour suppressor pathways involving p53 and p21CIP1, as well as p16INK4a and RB1 [21]. However, persistent senescent cells acquire a senescence-associated secretory phenotype (SASP), characterised by the chronic release of pro-inflammatory cytokines (IL-6, IL-8), growth factors, and proteases [22]. Over time, SASP fosters a microenvironment conducive to tumour initiation and metastasis by promoting extracellular matrix degradation, angiogenesis, and immune evasion [23]. Moreover, senescence escape—facilitated by the loss of TP53, RB1, or telomerase reactivation (TERT)—can give rise to highly malignant clones with severe genomic aberrations, as detailed in Table 1. Therefore, while senescence is initially protective, its dysregulation or escape significantly contributes to cancer progression in aged tissues. Given the multifactorial nature of cancer's adaptive landscape, single-target therapeutics are often inadequate [24]. Consequently, contemporary oncology increasingly embraces multitarget strategies that modulate several interconnected pathways simultaneously [25–27]. Polypharmacology improves efficacy, minimises resistance, and better captures the complexity of cancer biology—particularly in ageing-associated malignancies [28].

Advances in computational chemistry and structural bioinformatics have revolutionised the discovery of multitarget inhibitors. High-resolution crystal structures and algorithmic progress enable virtual screening across multiple targets *in-silico*, dramatically reducing experimental costs [29,30]. A typical computational drug discovery workflow begins with meticulous protein and ligand preparation, ensuring accurate geometries and protonation states. Subsequently, multitarget molecular docking predicts the optimal binding orientations and affinities of ligands within active or allosteric pockets of multiple proteins [31]. In this process, binding poses are evaluated by scoring functions and validated using known reference inhibitors [32]. Molecular interaction fingerprints (MIFs) are generated to gain deeper insights into atomic-level interactions. MIFs quantitatively encode hydrogen bonding, hydrophobic contacts, π–π stacking, and electrostatic interactions, allowing comparison of binding patterns across targets [33]. Such fingerprint analyses are invaluable for assessing whether a single molecule can establish stable and conserved interactions across diverse proteins—an essential property for multitarget inhibitors. At the quantum mechanical level, Density Functional Theory (DFT) calculations are utilised to explore the electronic structure of lead compounds [33,34]. By evaluating frontier molecular orbitals (HOMO–LUMO gap), molecular electrostatic potential (MEP), and dipole moment, DFT analysis elucidates the compound's chemical reactivity,

**Table 1. Target proteins used in the study, their main functions, and associated pathway categories.**

| PDB IDs | Proteins | Main Function | Category | References |
|---------|----------|---------------|----------|------------|
| 2YEX | Chk (Checkpoint kinase1) | DNA damage checkpoint kinase | DNA Damage Repair and Senescence Escape | [17] |
| 4HG7 | MDM2 (Mouse double minute 2 homolog) | DNA damage checkpoint kinase (p53 signalling) | DNA repair + growth signalling + apoptosis escape | [18] |
| 4JSX | mTOR (Mechanistic Target of Rapamycin) | Cell survival and growth signalling | DNA repair + growth signalling + apoptosis escape | [19] |
| 5DS3 | PARP-1 (Poly [ADP-ribose] polymerase 1) | Growth, metabolism, and survival regulator | DNA repair + growth signalling + apoptosis escape | [20] |

charge transfer potential, and stability—parameters that directly influence binding behaviour and redox balance within biological systems [33]. Complementing these, WaterMap analysis evaluates the thermodynamic role of water molecules at the binding interface. By mapping hydration sites and computing their enthalpic and entropic contributions, WaterMap identifies displaceable high-energy water molecules whose removal upon ligand binding enhances affinity [35]. This solvation insight helps rationalise binding energetics beyond traditional docking models, especially in systems where water-mediated hydrogen bonding is critical [36]. However, docking offers a largely static representation of molecular interactions. Therefore, molecular dynamics (MD) simulations capture the conformational dynamics of protein–ligand complexes over time under near-physiological conditions [37]. MD simulations provide time-resolved insights into structural stability, flexibility, and key residue fluctuations. Trajectory analyses—such as root mean square deviation (RMSD), root mean square fluctuation (RMSF), radius of gyration (Rg), and hydrogen bond occupancy—help elucidate the persistence of interactions and global stability of complexes [38,39]. From these equilibrated trajectories, binding free energy calculations (MM-PBSA or MM-GBSA) are performed to estimate the thermodynamic favourability of ligand binding by integrating van der Waals, electrostatic, polar, and nonpolar solvation energies [40]. These energetic parameters refine docking predictions and provide quantitative affinity measures under dynamic equilibrium.

In this study, we employed an integrated computational strategy to identify FDA-approved drugs with multitarget inhibitory potential against key regulators of DNA repair and senescence escape—Chk1 (2YEX), MDM2 (4HG7), mTOR (4JSX), and PARP-1 (5DS3) [17–20]. Using a comprehensive pipeline combining multitarget docking, molecular interaction fingerprinting, DFT, and WaterMap solvation profiling, we systematically evaluated the structural, dynamic, and thermodynamic determinants of drug–target interactions and MD simulations, and its trajectories for binding free energy analysis, This multidimensional approach enabled the identification of compounds exhibiting stable, energetically favourable, and electronically compatible binding across all targets. Our study demonstrates a rational and computationally driven framework for repurposing existing drugs as multitarget inhibitors capable of modulating DNA repair and senescence pathways—providing a promising route toward therapeutic intervention in ageing-associated cancers.

## 2 Methodology

In this study, we employed a comprehensive computational workflow to identify potential multitarget inhibitors against four key proteins involved in DNA damage repair and senescence escape mechanisms associated with ageing-related cancers. The overall pipeline included protein and ligand preparation, multitarget molecular docking, comparison with the control, DFT analysis, WaterMap solvation studies, MD simulations, and MM/GBSA binding free energy computations. The complete methodological flow is summarised in Fig 1 for clarity and proper understanding of the complex methods.

### 2.1 Protein structure data collection and preparation

Accurate selection and meticulous preparation of protein structures are essential for reliable molecular docking studies, as the quality of input structures directly affects the precision of the docking outcomes [26]. The biologically relevant proteins' crystallographic structures were selected and preprocessed to eliminate non-essential artefacts that could interfere with ligand binding. The four key proteins involved in DNA damage repair and senescence escape pathways—2YEX (Chk1 kinase), 4HG7 (MDM2 E3 ubiquitin ligase), 4JSX (mTOR kinase domain), and 5DS3 (PARP-1 DNA repair enzyme)—were retrieved from the RCSB Protein Data Bank (https://www.rcsb.org/) [17–20,41], and preparation was carried out using the Protein Preparation Workflow in Schrödinger Maestro v2024-4 [42]. All non-essential water molecules, heteroatoms, and redundant chains were removed. Missing side chains were reconstructed using Prime, and protonation states were optimised at physiological pH ($7.4 \pm 2$) [43]. The proteins were energy-minimised under the OPLS4 force field to ensure structural stability before grid generation and docking [44]. All protein structures were cleaned by retaining only the relevant chain. This process included three main stages: preprocessing,

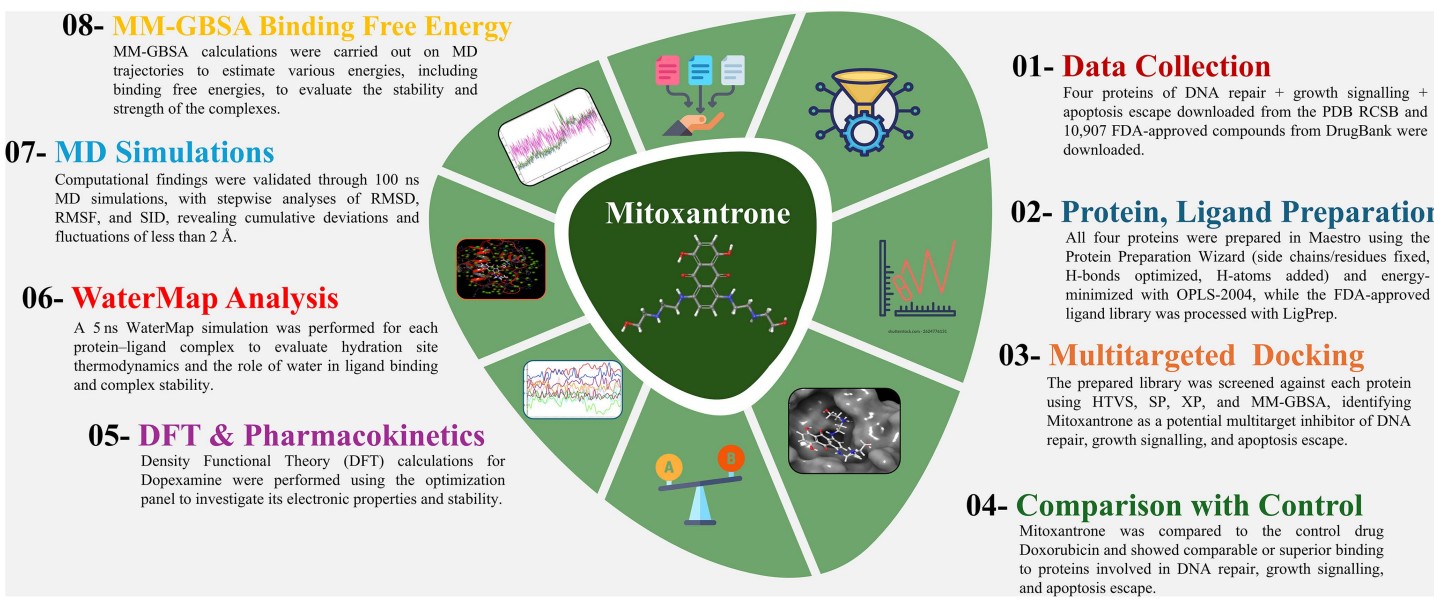

**Fig 1. Schematic of the computational workflow for the in-silico identification and validation of Mitoxantrone as a multitarget inhibitor of proteins implicated in ageing-associated cancers.** The pipeline consists of two main stages: **(I)** initial screening (steps 1-4), including data collection, preparation, multitargeted docking, and comparison to a control; and **(II)** detailed biophysical validation (steps 5-8), including DFT, WaterMap analysis, MD simulations, and MM-GBSA binding free energy calculations.

optimisation, and minimisation. Terminal capping was performed during preprocessing, missing side chains were added, and bond orders were assigned to chemical components. Hydrogens were replaced, disulfide bonds were created, and zero bond orders were applied to metal atoms. Missing loops were modelled using Prime, and protonation states were generated using Epik at physiological pH 7.24 ± 2 [42,43,45,46]. For optimisation, the system was refined by sampling water orientations and minimising hydrogens of altered species using PROPKA, while considering crystal symmetry [47]. In the minimisation step, atomic deviations were allowed up to 0.30 Å, the OPLS4 force field was applied, and all water molecules within 5.0 Å of native ligands were removed [44]. Following preparation, the final energy states of each structure were recorded, and Ramachandran plot analysis was performed to confirm structural quality and suitability for subsequent docking studies.

## 2.2 Ligand preparation

A library of Food and Drug Administration (FDA)-approved compounds was utilised to expedite the identification of safe, reproducible drug candidates [48]. A total of 2,648 approved drugs were downloaded from DrugBank (https://go.drugbank.com/) on 02 September 2025 [49]. Each compound was subjected to structural standardisation, protonation-state generation, and energy minimisation to ensure biologically relevant conformations suitable for molecular docking. Ligand preparation was performed using the LigPrep module in Schrödinger Maestro v2024-4 [50]. The following parameters were applied: a 500-atom limit, the OPLS4 force field, and generation of ionisation states at pH 7.0 ± 2 using the Epik module [46]. Desalting, tautomer generation, and stereoisomer enumeration were enabled, with a maximum of 32 stereoisomers per molecule, while retaining defined chiralities [50]. The final prepared library comprised 10,907 distinct 3D structures in SDF format that were ready for the docking studies. Additionally, Doxorubicin was downloaded and prepared using the same parameters as control ligands for comparative docking analysis [50].

## 2.3 Multitarget docking and pose refining

Molecular docking was carried out to predict the binding orientation and affinity of ligands within the active sites of the selected proteins. Receptor grids were generated using Schrödinger's Receptor Grid Generation tool for blind docking, encompassing the complete protein structures [51,52]. A scaling factor of 1.0 and a partial charge cutoff of 0.25 were applied. Binding regions were defined using residues around the centroid of selected residues, and grid box dimensions along the X, Y, and Z axes were manually adjusted to ensure appropriate coverage of potential binding sites [52]. Docking was executed via the Virtual Screening Workflow (VSW) in Maestro v2024-4. Prepared ligand libraries were loaded in the input tab, enabling compound redistribution for computational efficiency. The QikProp module was applied for pharmacokinetic and Lipinski's Rule-of-Five filtering [45,53]. Docking was conducted in three successive stages: High-Throughput Virtual Screening (HTVS), Standard Precision (SP), and Extra Precision (XP) modes. For each compound, four poses were generated [54]. The top-ranked poses from XP docking were subjected to Prime MM-GBSA (Molecular Mechanics Generalised Born Surface Area) calculations to estimate binding free energies [40]. Finally, docking results for all six protein targets were exported in CSV format for comparative evaluation. The same procedure was repeated for the control ligand Doxorubicin to assess the identified compound's relative binding strength and interaction quality in all four protein targets.

## 2.4 Molecular interaction fingerprints and comparative pharmacokinetics

MIFs were generated to characterise and compare the detailed interaction patterns between each ligand and its corresponding target protein. The Interaction Fingerprints tool in Schrödinger Maestro v2024-4 was used, aligning all protein–ligand complexes to account for structural and sequence variations [55]. Contacts were classified based on hydrogen bonding, hydrophobic, electrostatic, and van der Waals interactions. Non-interacting residues were removed, and the frequency and distribution of interacting residues were quantified to evaluate binding specificity. Pharmacokinetic properties, encompassing Absorption, Distribution, Metabolism, and Excretion (ADME), were analysed using the QikProp module in Maestro v2024-4 [45,53,55]. These results provided insight into the identified compound's physicochemical and biological suitability compared to standard reference ranges.

## 2.5 Density functional theory calculations

DFT computations were performed using the Jaguar module integrated within Schrödinger Maestro v2024-4 to optimise the geometry and evaluate the electronic properties of the selected compound [56,57]. The B3LYP-D3 hybrid functional with the 6-31G basis set was applied [58]. Dispersion corrections (D3 term) were included to account for van der Waals interactions critical in drug–receptor binding. Self-Consistent Field (SCF) spin treatment was set to automatic mode, with excited-state optimisations performed using Time-Dependent Density Functional Theory (TDDFT) for the first excited state and a maximum of 32 TDDFT interactions [59,60]. Energy convergence was set to $5 \times 10^{-5}$ eV, with a residual threshold of 0.01. Medium grid density and three-body dispersion corrections were enabled, and SCF convergence was accelerated using the Direct Inversion in the Iterative Subspace (DIIS) method [61,62]. A maximum of 100 optimisation steps was performed, employing a Schlegel guess for the initial Hessian. Key molecular descriptors, including HOMO, LUMO, and Natural Transition Orbitals (NTOs), were calculated. Solvation effects were modelled using the Polarizable Continuum Model (PCM) with water as the solvent, providing an implicit representation of solvent influence on the molecule [55].

## 2.6 WaterMap computations and hydration site characterisation

WaterMap analysis was conducted to elucidate the thermodynamic role of water molecules within the protein binding sites [35,36]. Using Schrödinger Maestro v2024-4, hydration sites were identified within 10 Å of the docked ligand [55]. Proteins were truncated appropriately, simulated using the OPLS4 force field, and water molecules were treated as explicit solvent. A 5 ns simulation was performed to achieve convergence [63]. Post-simulation analysis was conducted via the

WaterMap—Examine Results module to compute and visualise parameters including free energy (ΔG), enthalpy (ΔH), van der Waals interactions, entropy, and overlap factors [36]. All data were exported as CSV files and visual figures for comparative analysis across complexes [55].

## 2.7 Molecular dynamics simulation

To assess the dynamic stability and conformational behaviour of protein–ligand complexes, MD simulations were performed using the Desmond package integrated within Schrödinger Maestro v2024-4 (available from https://www.deshaw-research.com/) [55,63]. Simulation setup, production, and trajectory analysis were conducted in three stages. System setup was carried out using the System Builder tool, employing the TIP3P water model with an orthorhombic boundary and buffer dimensions of $10 \times 10 \times 10$ Å and the systems were neutralised by adding counterions [64]. The OPLS4 force field was used throughout [44]. Each complex underwent a 100 ns production run in the NPT ensemble at 300 K and 1.01325 bar, with trajectory frames recorded every 100 ps (total 1000 frames per simulation) [65]. RMSD, RMSF, and Simulation Interactions were analysed using the Simulation Interaction Diagram tool to assess structural stability and interaction persistence [55].

## 2.8 MM/GBSA binding free energy computations from MD trajectories

Binding free energy estimations were refined through MM-GBSA calculations to snapshots extracted from the MD trajectories [55,63]. These computations decomposed total binding energy into van der Waals, electrostatic, and solvation components, providing a detailed energetic profile of each complex. The thermal_mmgbsa.py script (from Schrödinger v2024-4) was executed on a Linux terminal using the following commands:

```
export SCHRODINGER=/opt/Schrodinger-VERSION/
$SCHRODINGER/run thermal_mmgbsa.py desmond_md_job_NAME-out.cms
```

The resulting free energy profiles quantified Mitoxantrone's energetic stability and overall affinity with each protein target, enabling precise comparison across all complexes [40,66].

# 3 Results

## 3.1 Prepared protein structure optimisation and validation

All protein structures were refined and energy-minimised using the Protein Preparation Workflow implemented in Schrödinger Suite (version 2024−4) to ensure accurate protonation states, correct bond orders, and optimal hydrogen-bonding networks before receptor grid generation. The minimisations were performed using the OPLS4 force field, which provides accurate parametrisation of bonded and non-bonded interactions for biomolecular systems. The prepared proteins included Chk1 kinase (PDB ID: 2YEX), MDM2 E3 ubiquitin ligase (PDB ID: 4HG7), mTOR kinase domain (PDB ID: 4JSX), and PARP-1 DNA repair enzyme (PDB ID: 5DS3). All minimisations were carried out to convergence at a heavy-atom root-mean-square deviation (RMSD) threshold of 0.30 Å, yielding energy-minimised, thermodynamically stable protein conformations suitable for molecular docking analyses. The total potential energy of each system—corresponding to the sum of bonded and non-bonded interactions—was found to be −351.483 kcal/mol (4HG7), −872.830 kcal/mol (5DS3), −1243.500 kcal/mol (2YEX), and −5231.600 kcal/mol (4JSX). The strongly negative values indicate that the structures reached a low-energy basin on the potential energy surface, consistent with physically realistic and stable conformations. As the systems were minimised in vacuo without molecular dynamics, the kinetic energy and system temperature were recorded as zero for all proteins, confirming that the obtained configurations represent static local minima rather than thermally fluctuating ensembles.

The bonded interaction energies—bond stretching, angle bending, and torsional terms—represent the internal geometric strain within the protein backbone and side chains after optimisation. The bond stretching energy, which quantifies

deviations of covalent bond lengths from their equilibrium distances, ranged from 52.003 kcal/mol in MDM2 (4HG7) to 649.567 kcal/mol in mTOR (4JSX). Higher stretching energy in mTOR reflects its large kinase domain with multiple secondary-structure elements and extended loops that require extensive relaxation. The angle bending energy, corresponding to deformation of bond angles, showed a similar trend, increasing with protein size and conformational complexity—from 210.934 kcal/mol in MDM2 to 3028.110 kcal/mol in mTOR. The torsional energy, arising from rotations about single bonds, was 141.599 kcal/mol (MDM2), 421.315 kcal/mol (Chk1), 445.687 kcal/mol (PARP-1), and 2661.360 kcal/mol (mTOR). This energy component is particularly relevant for flexible regions such as loops and side chains, where torsional strain must be resolved for accurate active-site geometry. The restraining energy (torsions) was zero for all proteins, as no external constraints were applied during the minimisation. The magnitude of these bonded terms reflects the extent of local structural rearrangements necessary to achieve sterically and energetically favourable geometries within each protein.

The non-bonded interaction energies, which include van der Waals (Lennard-Jones) and electrostatic terms, largely determine the folded-state stability and the internal packing of amino acid residues. These interactions dominate the overall potential energy of macromolecular systems. The 1,4 Lennard-Jones and 1,4 electrostatic interactions account for short-range interactions between atoms separated by three covalent bonds, typically involving side-chain contacts within secondary-structure motifs. These values ranged from 492.523 kcal/mol to 6430.880 kcal/mol for the 1,4 Lennard-Jones component and from 150.172 kcal/mol to 2067.240 kcal/mol for the 1,4 electrostatics. Larger positive values indicate stabilising local packing between adjacent residues and side chains. The global Lennard-Jones energy (–915.936 to –13454.100 kcal/mol) represents the cumulative attractive van der Waals forces between non-bonded atoms. These large negative values denote compact, well-packed tertiary structures where dispersion interactions contribute substantially to stabilisation. Similarly, the electrostatic energy, varying from –487.980 kcal/mol (MDM2) to –6948.070 kcal/mol (mTOR), arises from Coulombic interactions between charged and polar residues. The intense negative electrostatic energies observed, particularly in the kinase domains (2YEX and 4JSX), indicate optimal charge distributions and stabilised salt-bridge or dipole interactions across the active site and regulatory regions.

The workflow did not separately report explicit hydrogen-bond energies, as hydrogen-bonding effects are implicitly included within the overall non-bonded potential terms of the OPLS4 force field. Nevertheless, all systems' stable and negative potential energies suggest that intra-protein hydrogen bonds were preserved or optimised during relaxation, particularly in α-helices and β-sheets. All the energetic trends observed are consistent with each protein's structural complexity and molecular mass. mTOR (4JSX), being the largest kinase domain, exhibited the most negative total energy (–5231.600 kcal/mol), reflecting extensive non-bonded stabilisation from hydrophobic core packing and electrostatic complementarity. Conversely, MDM2 (4HG7) displayed the smallest magnitude of total energy (–351.483 kcal/mol), consistent with its smaller, globular fold. The uniformly negative total potential energies indicate that all prepared protein models attained geometrically stable, energetically minimised conformations that closely approximate their biologically relevant states. These optimised structures thus provide accurate receptor models for subsequent receptor grid generation and molecular docking experiments, ensuring that ligand-binding simulations are performed on physically realistic, low-energy protein conformations representative of their functional states in biological environments. Further, in Fig 2 and SS01, we have shown the prepared protein structures, energies, with Ramachandran plots to ensure the structure reliability.

## 3.2 Multitargeted Docking results & comparison with control inhibitors

Multitarget molecular docking and post-docking free energy refinement were performed using the Glide and Prime MM-GBSA modules of the Schrödinger Suite (v.2024−4). This analysis assessed the inhibitory potential of an FDA-approved DrugBank library, from which Mitoxantrone emerged as a top candidate and was compared against Doxorubicin (control compound). Four ageing-associated, cancer-relevant proteins were selected as targets: Checkpoint kinase 1 (Chk1, PDB ID: 2YEX), Murine double minute 2 (MDM2, PDB ID: 4HG7), mechanistic Target of Rapamycin kinase domain

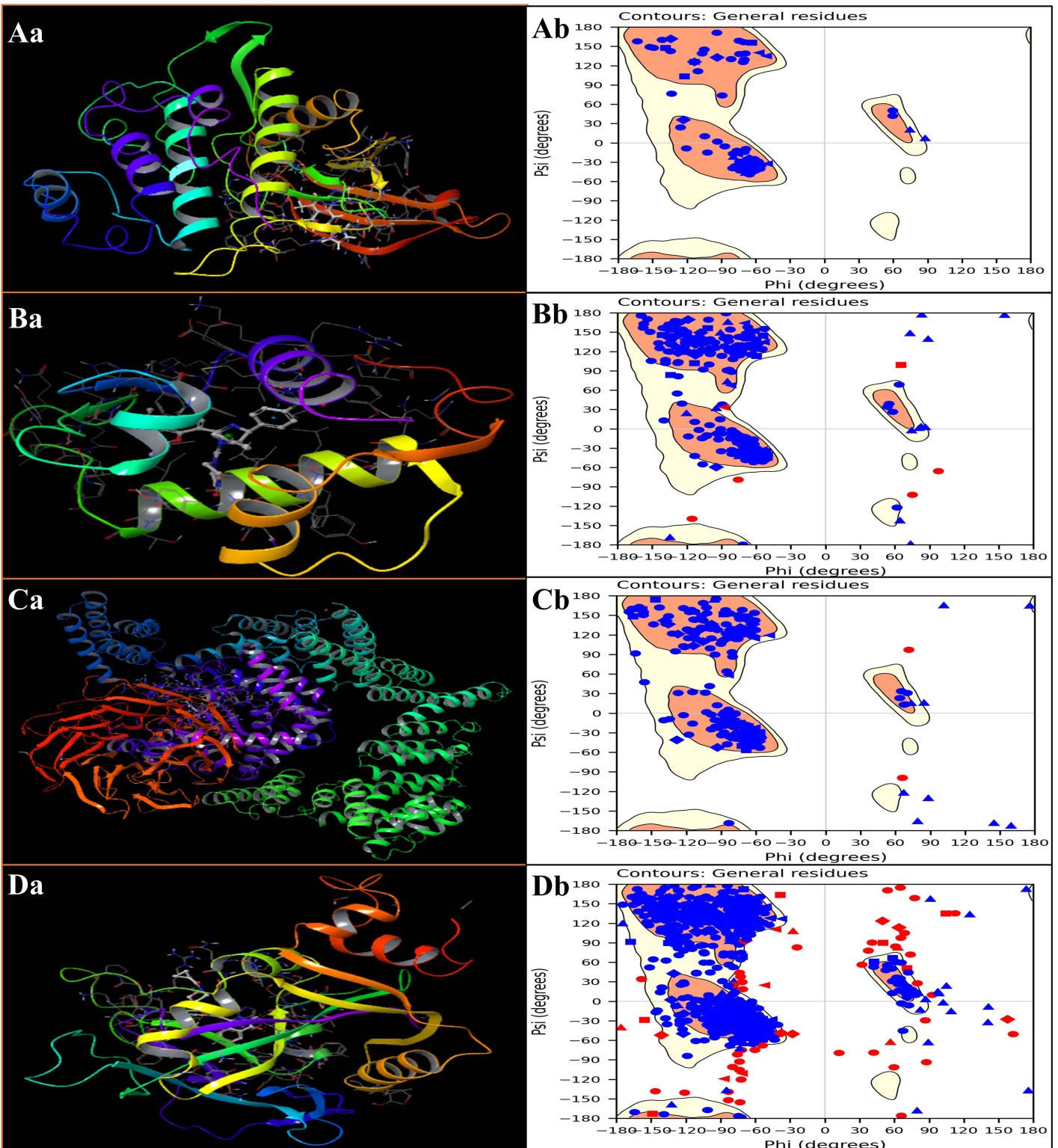

**Fig 2. The figure shows the prepared protein structures, their ligand binding sites, and the corresponding Ramachandran plots for the studied targets: A) Checkpoint kinase 1 (Chk1, PDB ID: 2YEX), B) MDM2 (Mouse double minute 2 homolog, PDB ID: 4HG7), C) mTOR (Mechanistic Target of Rapamycin kinase domain, PDB ID: 4JSX), and D) PARP-1 (Poly [ADP-ribose] polymerase 1 catalytic domain, PDB ID: 5DS3).**

(mTOR, PDB ID: 4JSX), and Poly(ADP-ribose) polymerase-1 catalytic domain (PARP-1, PDB ID: 5DS3). Detailed protocols and parameters for sorting and identifying the best candidates are provided in SS02, SS03, and SS04. The docking scores, MM-GBSA binding free energies (ΔG_bind), and ligand efficiencies are summarised in Table 2, and the two-dimensional ligand–receptor interaction diagrams are illustrated in Fig 3. Mitoxantrone exhibited strong binding affinities across all four protein targets, with docking scores ranging between –6.23 and –16.04 kcal/mol, and MM-GBSA binding free energies (ΔG_bind) from –49.19 to –85.14 kcal/mol (Table 2). The negative energy values indicate spontaneous and thermodynamically favourable interactions, reflecting optimal steric and electrostatic complementarity within the protein binding sites. For Chk1 kinase (2YEX), Mitoxantrone displayed the most favourable docking score (–16.04 kcal/mol) and the lowest MM-GBSA ΔG_bind (–85.14 kcal/mol), with a ligand efficiency of –2.661 kcal/mol/atom. The binding pose revealed multiple hydrogen bonds involving SER88 and GLU134 via its protonated amino groups, and additional interactions with CYS87, GLU85, and ASN135 through hydroxyl functionalities. These hydrogen-bonding networks stabilise the ligand within the ATP-binding site, while π–π and van der Waals contacts reinforce the ligand's anchoring, suggesting potential inhibition of Chk1 catalytic activity (Fig 3). In MDM2 (4HG7), Mitoxantrone exhibited a docking score of –6.23 kcal/mol and an MM-GBSA ΔG_bind of –49.19 kcal/mol, interacting through hydrogen bonds with VAL93 (via an $N^+H_2$ group) and GLN72 (via an OH group). Additionally, a π–cation interaction with HIS96 was observed, stabilising the ligand near the MDM2–p53 interface, indicating possible interference with p53 recognition and ubiquitination processes. Binding of Mitoxantrone to the mTOR kinase domain (4JSX) produced a docking score of –9.67 kcal/mol and an MM-GBSA ΔG_bind of –70.91 kcal/mol, reflecting strong van der Waals and electrostatic stabilisation within the active site. The ligand formed six hydrogen bonds with SER2165, GLN2167, VAL2240, CYS2243, and ARG2348. A π–π stacking interaction with TYR2225 further reinforced the planar anthracenedione ring orientation within the hydrophobic region of the catalytic pocket, highlighting its potential to modulate ATP-dependent phosphorylation in mTOR signalling (Fig 3). For PARP-1 (5DS3), Mitoxantrone demonstrated a docking score of –10.23 kcal/mol and an MM-GBSA ΔG_bind of –78.27 kcal/mol, accompanied by a ligand efficiency of –2.446 kcal/mol/atom. Seven hydrogen bonds were identified with SER864, ASN868, GLY863, SER904, TYR907, and GLU988, with a salt bridge forming between the ligand's protonated amine and GLU988 (Fig 3). This electrostatic contact provides substantial enthalpic stabilisation, suggesting high affinity toward the $NAD^+$-binding region, potentially impairing PARP-1's role in DNA repair and genomic stability.

The control compound Doxorubicin exhibited moderate binding across the same targets, with docking scores ranging from –5.21 to –8.05 kcal/mol and MM-GBSA ΔG_bind values between –39.99 and –66.04 kcal/mol (Table 2). Across all four proteins, Mitoxantrone consistently achieved lower (more negative) docking and binding energies, indicating stronger binding affinity and a higher degree of complementarity. For Chk1 (2YEX), Doxorubicin achieved a docking score of –7.21 kcal/mol and a ΔG_bind of –53.01 kcal/mol, forming four hydrogen bonds with THR14, CYS87, and GLH91 (Fig 3).

Table 2. Molecular docking scores (kcal/mol) and MM-GBSA binding energies (kcal/mol) for each target protein (PDB ID), including contributions from hydrogen bonds and vdW interactions.

| PDB ID | Reso-lution | Ligand | gridbox xcent | gridbox ycent | gridbox zcent | Docking Score | Prime Hbond | MMGBSA dG Bind | ligand efficiency |
|---|---|---|---|---|---|---|---|---|---|
| 2YEX | 1.3 | Mitoxantrone (Identified) | 1.954 | 37.493 | 10.976 | −16.044 | −151.6 | −85.14 | −2.661 |
| 4HG7 | 1.6 | | −23.734 | 7.129 | −14.023 | −6.23 | −51.48 | −49.19 | −1.537 |
| 4JSX | 3.5 | | 50.236 | −1.388 | −47.641 | −9.672 | −755.9 | −70.91 | −2.216 |
| 5DS3 | 2.6 | | −2.607 | 39.914 | 14.655 | −10.239 | −122.9 | −78.27 | −2.446 |
| 2YEX | 1.3 | Doxorubicin (Control) | 1.954 | 37.493 | 10.976 | −7.214 | −149 | −53.01 | −1.359 |
| 4HG7 | 1.6 | | −23.734 | 7.129 | −14.023 | −5.212 | −51.52 | −39.99 | −1.025 |
| 4JSX | 3.5 | | 50.236 | −1.388 | −47.641 | −8.053 | −754.1 | −66.04 | −1.693 |
| 5DS3 | 2.6 | | −2.607 | 39.914 | 14.655 | −5.382 | −121.3 | −47.43 | −1.216 |

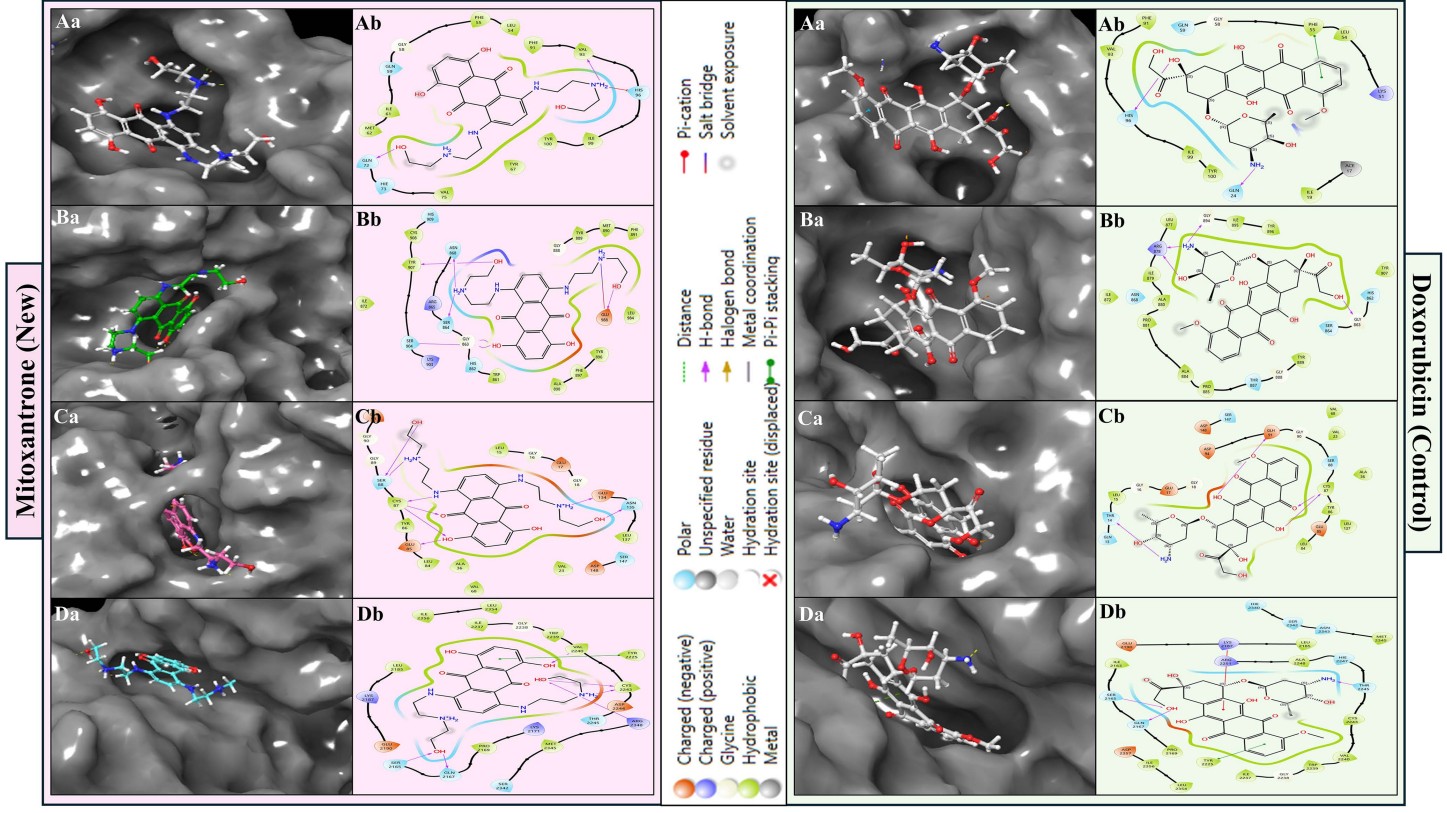

**Fig 3. The figure shows the docked poses in a) 3D and b) 2D Ligand Interaction Diagrams for the studied targets in complex with identified compound Mitoxantrone (light pink) and control drug Doxorubicin (sweet leaf; greenish), with a legend indicating bond and residues and interaction types.** The interaction of both compounds with protiens are shown as **A)** Checkpoint kinase 1 (Chk1, PDB ID: 2YEX), **B)** MDM2 (Mouse double minute 2 homolog, PDB ID: 4HG7), **C)** mTOR (Mechanistic Target of Rapamycin kinase domain, PDB ID: 4JSX), and **D)** PARP-1 (Poly [ADP-ribose] polymerase 1 catalytic domain, PDB ID: 5DS3).

Although these interactions are stabilising, the absence of additional electrostatic contacts limits their affinity compared to Mitoxantrone. In MDM2 (4HG7), Doxorubicin showed a weaker docking score (–5.21 kcal/mol) and ΔG_bind (–39.99 kcal/mol), interacting via hydrogen bonds with GLN24 and HIS96, and exhibiting a π–π stacking interaction with PHE55 (Fig 3). These limited interactions suggest that Doxorubicin's planar anthracycline structure does not engage the full hydrophobic groove involved in MDM2–p53 recognition. Binding to mTOR (4JSX) yielded a docking score of –8.05 kcal/mol and ΔG_bind of –66.04 kcal/mol, where Doxorubicin formed three hydrogen bonds with THR2245, SER2165, and GLN2167, in addition to π–π stacking with TYR2225 and LYS2187 (Fig 3). While these interactions are comparable in type to Mitoxantrone, their overall binding energies were less favourable due to weaker electrostatic contributions. For PARP-1 (5DS3), Doxorubicin displayed the least favourable interaction profile (docking score –5.38 kcal/mol, ΔG_bind –47.43 kcal/mol), forming four hydrogen bonds with ARG878, GLY894, and GLY863 (Fig 3). The lack of strong electrostatic or salt-bridge interactions, as seen with Mitoxantrone, accounts for its relatively lower binding affinity.

The overall binding affinity trend for Mitoxantrone followed the order: Chk1 (2YEX)> PARP-1 (5DS3) > mTOR (4JSX)> MDM2 (4HG7), which parallels the MM-GBSA ΔG_bind values (–85.14, –78.27, –70.91, and –49.19 kcal/mol, respectively). Doxorubicin followed a similar trend but with less favourable binding energies across all targets. The superior ligand efficiency values of Mitoxantrone (–2.661 to –1.537 kcal/mol/atom) relative to Doxorubicin (–1.359 to –1.025 kcal/mol/atom) further confirm its better binding energy normalised per heavy atom, indicating a more efficient utilisation

of molecular interactions. Mechanistically, Mitoxantrone's polyhydroxylated anthracenedione scaffold enables multiple hydrogen bonds and electrostatic interactions, while its planar aromatic core allows π–π stacking with aromatic residues, collectively conferring stronger and more versatile binding. This multitarget profile supports its potential as a polypharmacological inhibitor capable of modulating key cancer-related signalling pathways, including DNA repair (PARP-1), cell-cycle checkpoint regulation (Chk1), mTOR-mediated growth signalling, and MDM2-mediated p53 degradation. Thus, Mitoxantrone demonstrates a broader and more energetically favourable binding spectrum than Doxorubicin, making it a promising candidate for multitarget therapeutic intervention in ageing-associated malignancies.

### 3.3 Analysis of molecular interaction fingerprints

A comprehensive MIFs analysis was performed on the Mitoxantrone–target complexes to elucidate the residue-specific interaction patterns contributing to ligand stabilisation. The MIF data were generated by profiling the amino acid residues participating in hydrogen bonding, hydrophobic contacts, electrostatic, and π-interactions within 4 Å of the ligand atoms. The frequency and distribution of interacting residues with their counts were 6GLY, 6VAL, 5GLU, 5LEU, 4ALA, 3ASP, 3TYR, 2ARG, 2CYS, 2HIS, 2PHE, 2PRO, 2SER, 1ASN, 1ILE, 1LYS, 1THR, and 1TRP. The MIF profile revealed that Glycine (GLY) and Valine (VAL) were the most frequently involved residues, each contributing six contacts across the four protein–ligand complexes (Fig 4). The dominance of these residues reflects their prevalence within flexible loop and β-turn regions surrounding active sites. Glycine, the smallest amino acid, often permits local conformational flexibility, enabling optimal spatial accommodation of Mitoxantrone within the catalytic clefts. Valine, a nonpolar residue, contributes hydrophobic stabilisation via van der Waals contacts with the planar anthracenedione core, facilitating compact stacking within

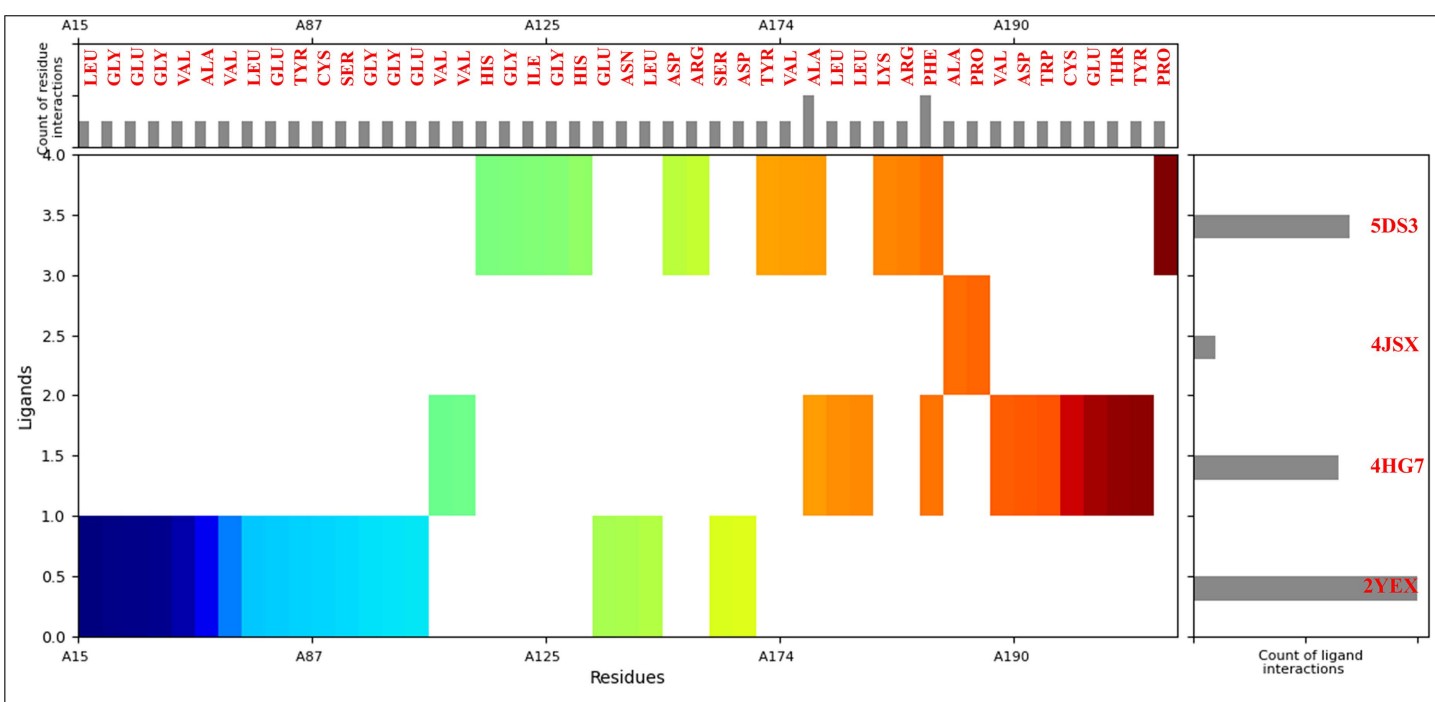

**Fig 4. The figure shows the Molecular Interaction Fingerprints (MIFs) of docked poses of identified compound Mitoxantrone and Checkpoint kinase 1 (Chk1, PDB ID: 2YEX), MDM2 (Mouse double minute 2 homolog, PDB ID: 4HG7), mTOR (Mechanistic Target of Rapamycin kinase domain, PDB ID: 4JSX), and PARP-1 (Poly [ADP-ribose] polymerase 1 catalytic domain, PDB ID: 5DS3).** The count of interacting residues is shown as a bar on top, while the count of ligand interactions is shown at the right side of the plot.

the binding pocket. Glutamic acid (GLU) and Leucine (LEU), each observed in five interactions, further reinforce electrostatic and hydrophobic complementarity. The negatively charged carboxylate groups of GLU residues form salt bridges or hydrogen bonds with the protonated amino groups of Mitoxantrone, providing enthalpically favourable electrostatic stabilisation. Conversely, LEU residues—frequently located at the periphery of hydrophobic pockets—stabilise the aromatic rings of Mitoxantrone through dispersion forces, supporting the ligand's spatial anchoring and shape complementarity. Alanine (ALA), observed four times, represents small hydrophobic contacts that modulate ligand positioning within the hydrophobic patches of the protein cores. Its moderate frequency suggests a role in providing subtle steric support without contributing to directional bonding. With three occurrences each, aspartic acid (ASP) and Tyrosine (TYR) participate in key polar and π-mediated interactions. ASP residues frequently form charge-assisted hydrogen bonds with Mitoxantrone's hydroxyl or amine functionalities, enhancing binding specificity. TYR, possessing both aromatic and polar characteristics, mediates π–π stacking and hydrogen bonding interactions, particularly stabilising the ligand's planar anthracenedione moiety via π–π stacking and hydrogen bonds through its hydroxyl group. Moderately represented residues such as Arginine (ARG), Cysteine (CYS), Histidine (HIS), Phenylalanine (PHE), Proline (PRO), and Serine (SER) (each appearing twice) contribute diverse noncovalent interactions (Fig 4). ARG residues establish salt bridges and π–cation interactions with the aromatic systems of Mitoxantrone, while CYS residues often participate in weak hydrogen bonding or nonpolar interactions, especially within catalytic regions rich in thiol functionality (e.g., Chk1). HIS residues contribute π–π stacking and hydrogen bonding, consistent with their role as catalytic residues in kinase and oxidoreductase domains. PHE residues stabilise the ligand via strong π–π stacking, enhancing aromatic complementarity within the hydrophobic core. PRO residues contribute through backbone rigidity, maintaining the conformational integrity of loops involved in ligand recognition, whereas SER residues mediate hydrogen bonds via their hydroxyl groups, reinforcing polar complementarity at solvent-exposed binding interfaces.

Residues appearing once—Asparagine (ASN), Isoleucine (ILE), Lysine (LYS), Threonine (THR), and Tryptophan (TRP)—represent localised but structurally significant interactions. ASN and THR form directional hydrogen bonds stabilising polar termini of the ligand, while LYS engages in electrostatic interactions with oxygen atoms of carbonyl or hydroxyl groups. ILE contributes to hydrophobic stabilisation at the boundary of lipophilic cavities. TRP participates through π–π stacking or edge-to-face interactions, often contributing to aromatic stabilisation within the catalytic pocket. The MIF distribution underscores the dual electrostatic–hydrophobic nature of Mitoxantrone binding. The high prevalence of polar acidic (GLU, ASP) and basic (ARG, LYS, HIS) residues suggests substantial electrostatic stabilisation, consistent with the molecule's zwitterionic features and strong hydrogen-bonding capacity. Simultaneously, the abundance of nonpolar aliphatic (VAL, LEU, ALA, ILE) and aromatic (TYR, PHE, TRP) residues reflects extensive hydrophobic and π-mediated complementarity (Fig 4). This balanced interaction pattern indicates that Mitoxantrone achieves strong binding through synergistic electrostatic and hydrophobic stabilisation, rather than relying on a single dominant interaction type. Such a mixed-interaction fingerprint aligns with the compound's ability to engage structurally diverse protein targets—Chk1, MDM2, mTOR, and PARP-1—suggesting a mechanistically consistent polypharmacological binding behaviour. The MIF analysis demonstrates that Mitoxantrone's interaction network involves a heterogeneous but cooperative ensemble of residues—predominantly glycine, valine, glutamate, and leucine—contributing flexibility and affinity. This multi-residue engagement underpins its strong binding energies and multitarget inhibitory potential against ageing-associated oncogenic proteins. Furthermore, a detailed view is shown in Fig 4.

## 3.4 Analysis of DFT and pharmacokinetics

To characterise the electronic structure, frontier molecular orbitals, and reactivity distribution of Mitoxantrone (DrugBank ID: DB01204), a time-dependent density functional theory (TD-DFT) calculation was performed using the B3LYP-D3 hybrid functional with a 6-31G** basis set. The computation was conducted under gas-phase conditions with a spin multiplicity of 1, corresponding to a closed-shell singlet ground state, and the geometry convergence category was 2, indicating

a well-converged optimised structure with minimal residual forces. The final optimised energy of Mitoxantrone was found to be –1526.3209 a.u., while the gas-phase ground-state energy reached –1526.3885 a.u., confirming the thermodynamic stability of the optimised structure. The target excited-state energy was identical to the final total energy (–1526.3209 a.u.), suggesting that the molecule exhibits minimal structural distortion upon excitation, consistent with a rigid π-conjugated anthracenedione scaffold. The Frontier Molecular Orbital and Electronic Excitation Analysis, which includes the Highest Occupied Molecular Orbital (HOMO) and Lowest Unoccupied Molecular Orbital (LUMO) energies, were calculated as –0.3857 eV and –0.3115 eV, respectively. The resulting HOMO–LUMO energy gap (ΔE) of 0.0742 eV indicates a relatively narrow electronic band gap, characteristic of high molecular polarisability and enhanced electron delocalisation across the anthracenedione core (Fig 5, SS05). Such a narrow gap supports the compound's ability to participate in efficient charge-transfer interactions with electron-deficient protein residues, particularly within redox-active or π-rich catalytic pockets such as Chk1 kinase and PARP-1. The first singlet–singlet electronic excitation energy was computed at 1.838 eV with a corresponding oscillator strength of 0.1885, signifying a moderately allowed electronic transition dominated by π–π character. This transition, likely localised on the planar quinone and aromatic regions, correlates with the optical activity observed in anthracenedione derivatives and underpins the molecule's charge-transfer capability in biological recognition processes.

The Average Local Ionisation Energy (ALIE) analysis evaluated the molecule's localised electron detachment tendencies and potential electrophilic reactivity. The mean ALIE was determined to be 410.6 kcal/mol, with a minimum ALIE of 318.98 kcal/mol and a maximum ALIE of 524.94 kcal/mol, yielding an average absolute deviation of approximately 28.97

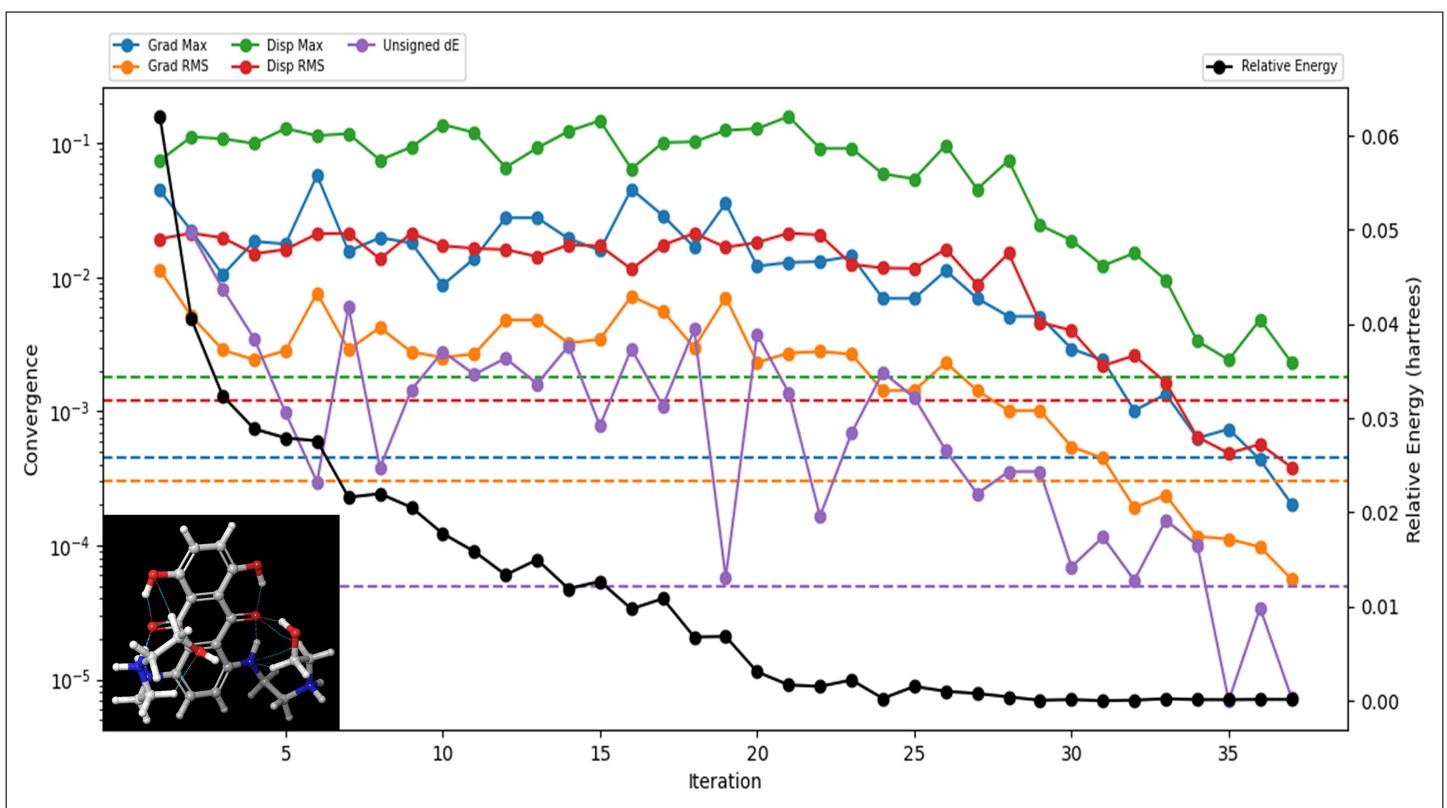

**Fig 5. The figure shows the Density Functional Theory (DFT) Results of the identified compound Mitoxantrone, where the relative energy is shown along with all other energies.**

kcal/mol. The total ALIE variance was 1405.43 (kcal/mol)$^2$, indicating a moderately heterogeneous electronic density distribution across the molecule. The lower ALIE regions (≈319 kcal/mol) correspond to electron-rich areas localised around the quinone carbonyl oxygens and amino substituents, reflecting sites of potential electrophilic attack or hydrogen-bond acceptor capability. Conversely, higher ALIE zones (≈525 kcal/mol) were primarily concentrated on the aromatic rings and peripheral hydroxyl groups, denoting regions of lower ionisation propensity but high π-electron stability. The balance between these regions implies that Mitoxantrone possesses a mixed electronic character — simultaneously nucleophilic (via carbonyl oxygen lone pairs) and electrophilic (via π-deficient aromatic centres) — which may enhance its adaptability in binding to heterogeneous protein microenvironments. The electrostatic potential (ESP) parameters further illustrate the polarity and charge distribution within the molecule. The mean ESP was 123.9 kcal/mol, ranging from 68.68 kcal/mol to a maximum of 183.18 kcal/mol, with an ESP local polarity of 18.64 kcal/mol. The total ESP variance of 554.69 (kcal/mol)$^2$ indicates moderate polar heterogeneity across the molecular surface. Positive ESP regions were predominantly located near the protonated amine groups, serving as potential hydrogen-bond donors, while negative ESP areas were concentrated around the carbonyl and hydroxyl oxygens, consistent with hydrogen-bond acceptor roles in protein–ligand recognition. The overall ESP and ALIE balance of zero indicates that the molecule maintains electronic neutrality and symmetrical charge distribution, favouring stable noncovalent interactions in binding pockets with polar and hydrophobic residues. These findings align with the observed docking profiles against Chk1, MDM2, mTOR, and PARP-1, where electrostatic complementarity and charge-assisted hydrogen bonding played dominant roles. The DFT analysis confirms that Mitoxantrone (DB01204) exhibits excellent electronic stability, high polarisability, and balanced charge distribution, supporting its capacity for multitarget binding. The narrow HOMO–LUMO gap and moderate oscillator strength point to a readily excitable electronic system conducive to π–π stacking, charge transfer, and redox adaptability. The localised ALIE minima and polar ESP regions reinforce the molecule's dual capacity to act as a hydrogen-bond donor and acceptor, facilitating stable interactions with diverse protein environments. Furthermore, Fig 5 is plotted to visualise all the energies properly.

The pharmacokinetic and ADME (absorption, distribution, metabolism, and excretion) characteristics of Mitoxantrone (DB01204) and the control drug Doxorubicin were evaluated using QikProp (Schrödinger Release 2024−4). The computed physicochemical and pharmacokinetic descriptors are shown in Table 3. The analysis provided detailed insights into both compounds' absorption potential, lipophilicity, membrane permeability, and plasma protein binding, elucidating their relative suitability for systemic administration and multitarget bioavailability. Mitoxantrone exhibited a molecular weight (MW) of 444.49 g·mol$^{-1}$, whereas Doxorubicin was heavier at 543.53 g·mol$^{-1}$. Both fall within the drug-like range (<600 g·mol$^{-1}$) for orally active compounds. Mitoxantrone contained 10 oxygen and nitrogen atoms (#NandO) compared with 12 in Doxorubicin, consistent with its relatively lower polarity. The number of rotatable bonds (#rotor) was slightly higher for Mitoxantrone (16) than for Doxorubicin (11), suggesting greater conformational flexibility that may facilitate binding adaptability across multiple protein targets. The polar surface area (PSA) was 183.24 Å$^2$ for Mitoxantrone and 215.69 Å$^2$ for Doxorubicin, exceeding the optimal threshold (~140 Å$^2$) for high passive membrane permeability. However, Mitoxantrone's comparatively smaller PSA implies moderately improved membrane penetration. Mitoxantrone was predicted to undergo 11 metabolic reactions (#metab), slightly higher than Doxorubicin (9), implying higher susceptibility to oxidative metabolism, possibly due to its multiple hydroxyl and amine groups exposed to cytochrome P450 oxidation. The globularity index (glob) of 0.835 for Mitoxantrone versus 0.805 for Doxorubicin indicates a slightly more compact molecular topology, which may aid target recognition and reduce non-specific interactions. The predicted octanol/water partition coefficient (QPlogPo/w) for Mitoxantrone was 0.366, indicating a balanced hydrophilic–lipophilic character. In contrast, Doxorubicin displayed a slightly negative value (–0.453), reflecting higher hydrophilicity. These findings align with the calculated aqueous solubility (QPlogS) values of –0.433 for Mitoxantrone and –2.454 for Doxorubicin, demonstrating that Mitoxantrone possesses approximately one order of magnitude greater solubility in aqueous medium. Similarly, the calculated intrinsic solubility (CIQPlogS) of Mitoxantrone (–2.932) was more favourable than that of Doxorubicin (–4.022). The polarizability (QPpolrz) was lower for Mitoxantrone (37.90 Å$^3$) relative to Doxorubicin (47.27 Å$^3$), suggesting reduced dispersion

**Table 3. Predicted pharmacokinetic and physicochemical parameters of Mitoxantrone and Doxorubicin computed using QikProp (Schrödinger v2024-4).**

| Title | Mitoxantrone | Doxorubicin | Title | Mitoxantrone | Doxorubicin |
|---|---|---|---|---|---|
| #NandO | 10 | 12 | Jm | 0 | 0 |
| #acid | 0 | 0 | PISA | 155.284 | 144.379 |
| #amide | 0 | 0 | PSA | 183.235 | 215.686 |
| #amidine | 0 | 0 | %HumanOralAbsorption | 19.199 | 0 |
| #amine | 2 | 1 | QPPCaco | 1.485 | 2.355 |
| #in34 | 0 | 0 | QPPMDCK | 0.531 | 0.79 |
| #in56 | 14 | 24 | QPlogBB | −2.244 | −2.974 |
| #metab | 11 | 9 | QPlogHERG | −6.395 | −6.094 |
| #nonHatm | 32 | 39 | QPlogKhsa | −0.384 | −0.601 |
| #noncon | 0 | 9 | QPlogKp | −8.65 | −7.888 |
| #ringatoms | 14 | 24 | QPlogPC16 | 14.408 | 16.339 |
| #rotor | 16 | 11 | QPlogPo/w | 0.366 | −0.453 |
| #rtvFG | 0 | 1 | QPlogPoct | 24.488 | 31.876 |
| #stars | 3 | 2 | QPlogPw | 16.411 | 24.216 |
| CNS | −2 | −2 | QPlogS | −0.433 | −2.454 |
| HumanOralAbs | 1 | 1 | QPpolrz | 37.897 | 47.274 |
| RuleOfFive | 1 | 3 | SASA | 697.093 | 778.721 |
| RuleOfThree | 2 | 2 | SAamideO | 0 | 0 |
| ACxDN^.5/SA | 0.0284037 | 0.0424976 | SAfluorine | 0 | 0 |
| CIQPlogS | −2.932 | −4.022 | WPSA | 0 | 0 |
| EA(eV) | 1.261 | 1.234 | accptHB | 9.9 | 14.8 |
| FISA | 276.079 | 318.546 | dip^2/V | 0.0434769 | 0.0225548 |
| FOSA | 265.73 | 315.797 | dipole | 7.577 | 5.772 |
| IP(eV) | 7.871 | 8.963 | donorHB | 4 | 5 |
| mol MW | 444.486 | 543.526 | glob | 0.8349808 | 0.8054716 |
| volume | 1320.393 | 1477.069 | Type | small | small |

interactions but potentially enhanced predictability of its binding kinetics. Both compounds exhibited low predicted human intestinal absorption, reflected by a Human Oral Absorption score of 1 (on a scale of 1–3) and a CNS activity score of −2, indicating poor blood–brain barrier (BBB) permeability and limited central nervous system exposure (Table 3). The percentage human oral absorption was 19.2% for Mitoxantrone and 0% for Doxorubicin, indicating that Mitoxantrone exhibits a measurable, though limited, degree of oral bioavailability, whereas Doxorubicin's systemic exposure following oral administration would be negligible. Predicted Caco-2 cell permeability (QPPCaco) and MDCK cell permeability (QPPMDCK) values for Mitoxantrone were 1.485 nm·s$^{-1}$ and 0.531 nm·s$^{-1}$, respectively, while Doxorubicin showed 2.355 nm·s$^{-1}$ and 0.79 nm·s$^{-1}$, indicating that Doxorubicin may traverse epithelial membranes slightly faster under passive diffusion conditions. However, both values are considerably below the ideal permeability threshold (typically >25 nm·s$^{-1}$), confirming that both molecules would rely on carrier-mediated or active transport mechanisms rather than passive diffusion. The predicted blood–brain barrier partition coefficient (QPlogBB) values of −2.244 for Mitoxantrone and −2.974 for Doxorubicin reaffirm their poor BBB permeability, which is advantageous in oncology, reducing off-target neurotoxicity. The predicted plasma protein binding (QPlogKhsa) values of −0.384 for Mitoxantrone and −0.601 for Doxorubicin suggest moderate to high serum albumin affinity, with Mitoxantrone being slightly less bound, thereby increasing its fraction of free, pharmacologically active drug. The HERG potassium channel inhibition potential (QPlogHERG), a predictor of cardiotoxicity,

was –6.395 for Mitoxantrone and –6.094 for Doxorubicin. Since values < –5.0 indicate possible HERG inhibition risk, both compounds show potential cardiotoxic tendencies, with Doxorubicin exhibiting marginally greater risk — consistent with its clinically established cardiotoxicity profile (SS06). Thus, Mitoxantrone demonstrates a slightly improved safety margin at the predicted level.

According to Lipinski's Rule of Five, Mitoxantrone violated only one rule, whereas Doxorubicin violated three, primarily due to excessive hydrogen-bond donors (5), acceptors (14.8), and high PSA (>200 Å²) (Table 3). This suggests that Mitoxantrone possesses superior overall oral drug-likeness. Both compounds passed the Rule of Three (score = 2), which indicates suitability for fragment-based lead optimisation. Mitoxantrone displayed 4 hydrogen-bond donors and 9.9 acceptors, whereas Doxorubicin had 5 donors and 14.8 acceptors, further supporting the control drug's higher hydrophilicity and reduced permeability. The dipole moment was 7.577 Debye for Mitoxantrone versus 5.772 Debye for Doxorubicin, implying stronger molecular polarity and potential for more directional electrostatic interactions in protein-binding pockets, particularly those rich in charged residues such as Asp, Glu, or Lys. Overall, the QikProp pharmacokinetic analysis demonstrates that Mitoxantrone exhibits a more favourable balance of solubility, lipophilicity, and drug-likeness than Doxorubicin, though both share limited oral absorption and low CNS permeability. Mitoxantrone's moderate hydrophobicity (QPlogPo/w = 0.366) and higher aqueous solubility (QPlogS = –0.433) suggest enhanced systemic availability and reduced aggregation potential compared with the control (SS06). Although Doxorubicin shows slightly higher passive permeability, its excessive polarity, higher molecular weight, and poor solubility collectively restrict its oral bioavailability and contribute to its known dose-limiting cardiotoxicity. Thus, these ADME predictions indicate that Mitoxantrone possesses a balanced pharmacokinetic and physicochemical profile, with potentially improved pharmacological manageability.

### 3.5 Analysis of WaterMap computations

The WaterMap simulations were performed to elucidate the thermodynamic contributions of hydration sites around the ligand-binding pockets of Mitoxantrone in complex with four protein targets: 4HG7, 5DS3, 2YEX, and 4JSX. Both three-dimensional (3D) and two-dimensional (2D) representations of the hydration patterns are illustrated in Fig 6. Panels Aa–Da represent the overall 3D hydration environments, whereas Ab–Db correspond to the detailed 2D residue interaction maps. The analysis provides insight into favourable and unfavourable water clusters, their displacement upon ligand binding, and the enthalpic–entropic balance driving complex stabilisation. In the 4HG7-Mitoxantrone complex, the Water-Map identified a highly ordered network of hydration sites within the hydrophilic binding pocket. Key interacting residues included GLY6, VAL6, GLU5, LEU5, ALA4, ASP3, TYR3, ARG2, CYS2, HIS2, and PHE2, consistent with the MIF profile. The 2D WaterMap (Ab) revealed several displaced high-energy hydration sites adjacent to ASP3, GLU5, and TYR3, indicating enthalpically unfavourable water molecules replaced by Mitoxantrone's hydroxyl and amino substituents. Two strong hydrogen bonds were observed between Mitoxantrone and GLU5 and ASP3, while π–π stacking interactions with TYR3 stabilised the aromatic anthracenedione core. Hydrophobic residues (VAL6, LEU5, and ALA4) formed a nonpolar microenvironment contributing to favourable entropy upon water displacement (Fig 6, SS07). The overall hydration reorganisation suggests that binding is predominantly entropy-driven, with partial enthalpic compensation from directional H-bonds. The 5DS3-Mitoxantrone complex displayed a mixed polar–hydrophobic cavity dominated by residues ASP3, GLU5, ARG2, TYR3, HIS2, and LEU5. The WaterMap (Bb) showed the presence of several structured hydration sites near ASP3 and ARG2, which were displaced upon ligand docking. These displacements correspond to high-enthalpy water molecules (ΔG ≈ +2–3 kcal/mol), positively contributing to the binding free energy. Mitoxantrone formed bidentate hydrogen bonds with GLU5 and ASP3, and additional salt-bridge interactions were observed with ARG2, stabilising the ligand orientation within the catalytic groove. The π–cation interaction between the protonated amine of Mitoxantrone and HIS2 further contributed to electrostatic complementarity. The displacement of structured water from hydrophobic regions around LEU5 and VAL6 enhanced entropic gain, indicating that hydration displacement in 5DS3 is both enthalpically and entropically favourable (Fig 6, SS07).

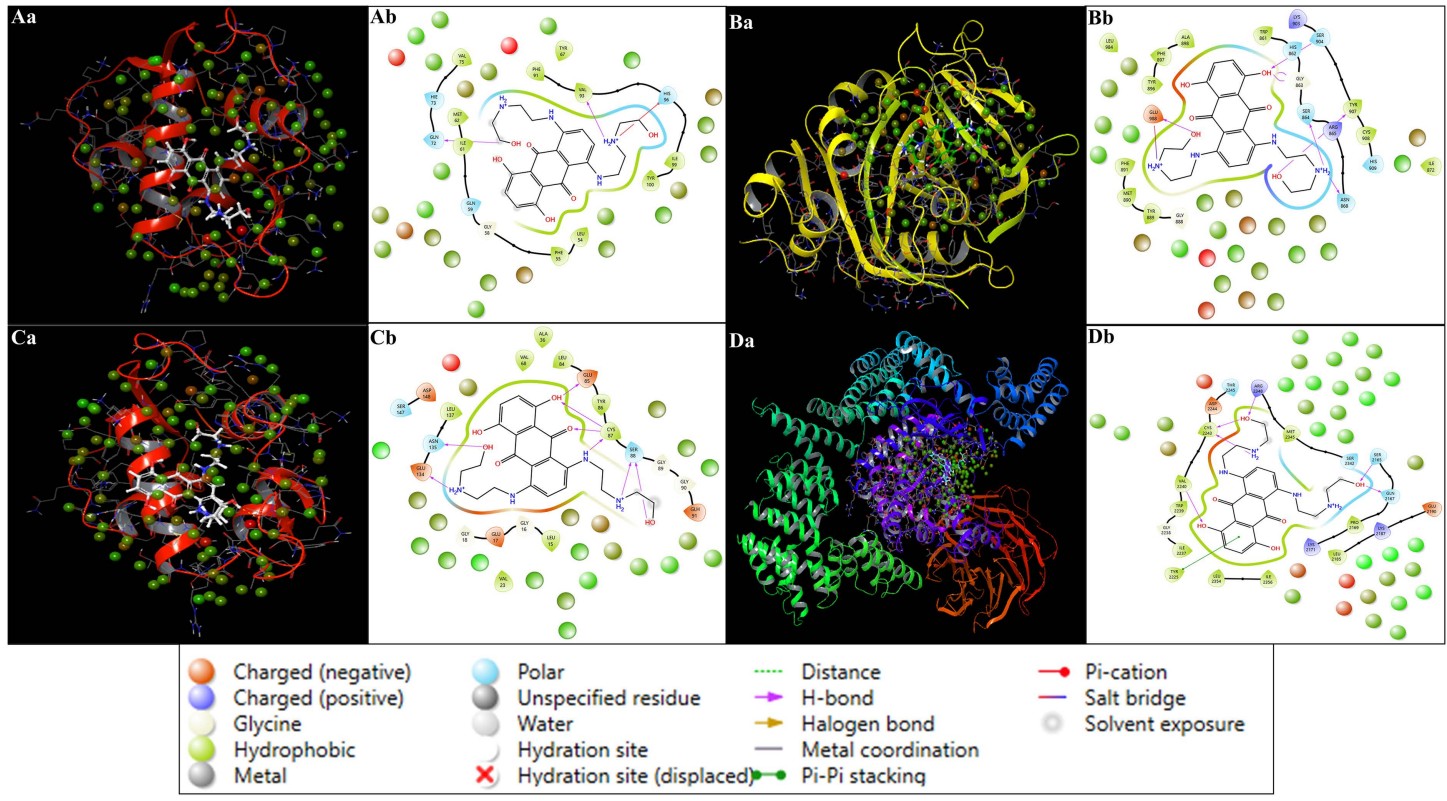

**Fig 6. WaterMap analysis of Mitoxantrone complexes with Mitoxantrone and Checkpoint kinase 1 (Chk1, PDB ID: 2YEX), MDM2 (Mouse double minute 2 homolog, PDB ID: 4HG7), mTOR (Mechanistic Target of Rapamycin kinase domain, PDB ID: 4JSX), and PARP-1 (Poly [ADP-ribose] polymerase 1 catalytic domain, PDB ID: 5DS3) showing 3D hydration (Aa–Da) and corresponding 2D residue interaction maps (Ab–Db).** The colour coding denotes residue properties and interaction types as indicated in the legend.

The 2YEX-Mitoxantrone structure exhibited a predominantly hydrophilic pocket with clusters of polar and charged residues, including GLU5, ASP3, ARG2, SER2, and HIS2, surrounded by hydrophobic residues VAL6, LEU5, and ALA4. The 2D WaterMap (Cb) displayed multiple hydration sites overlapping Mitoxantrone's hydroxyl and amine substituents, particularly near ASP3 and GLU5 (Fig 6, SS07). These sites corresponded to enthalpically unfavourable but entropically constrained water molecules, which enhanced the ligand's binding affinity upon displacement. The π–π stacking with PHE2 and TYR3 contributed to stabilising dispersion forces, while hydrogen bonding with SER2 reinforced polar complementarity. Several hydration sites were classified as "displaced" (red crosses in the 2D map), indicating efficient expulsion of high-energy water clusters upon ligand binding. This suggests that the Mitoxantrone–2YEX interaction benefits primarily from enthalpic stabilisation, resulting in a tighter, more specific binding orientation. In the 4JSX-Mitoxantrone complex, WaterMap analysis revealed a deep hydrophobic channel lined with LEU5, VAL6, ALA4, and PHE2, surrounded by polar residues GLU5, TYR3, and ARG2. The 2D WaterMap (Db) identified multiple displaced hydration sites adjacent to GLU5 and TYR3, consistent with ligand-induced desolvation in these polar pockets (Fig 6). The π–π stacking between the anthracenedione ring of Mitoxantrone and TYR3/PHE2 provided strong van der Waals stabilisation, while hydrogen bonds with GLU5 and ARG2 reinforced electrostatic complementarity. Notably, several hydration sites were retained near polar residues (HIS2 and SER2), suggesting a residual solvation layer that stabilises the local dielectric environment. The balance between displaced and retained hydration sites implies that binding is moderately enthalpy-driven, with contributions from hydrophobic desolvation and polar hydrogen-bond formation. Across all four complexes, WaterMap consistently

identified the reorganisation of high-energy hydration sites as a major contributor to the binding thermodynamics of Mitoxantrone. The recurrent involvement of GLU5, ASP3, TYR3, and ARG2 in hydrogen bonding or salt-bridge formation highlights their critical role in anchoring the ligand. Hydrophobic residues (VAL6, LEU5, ALA4, and PHE2) facilitated the displacement of unfavourable water molecules, contributing to the entropy-driven binding component. The WaterMap data suggest that Mitoxantrone binding is a cooperative process, driven by the favourable expulsion of high-energy water clusters and formation of specific hydrogen-bonding and π–stacking interactions. The hydration-site analysis complements the molecular docking and MIF results, providing a detailed thermodynamic rationale for Mitoxantrone's high binding affinity and structural adaptability.

### 3.6  Analysis of molecular dynamics simulation

MD simulations were performed using the Desmond module of the Schrödinger suite to elucidate the structural stability and interaction dynamics of mitoxantrone complexes with the target proteins bearing PDB IDs 4HG7, 5DS3, 2YEX, and 4JSX. The systems were solvated in an orthorhombic box employing the TIP3P water model, maintaining a buffer distance of 10 Å around the complex, and neutralised with counterions to simulate physiological conditions for 100 ns under an NPT ensemble at 300 K. Post-simulation evaluations included the calculation of deviation, RMSF, and bonding occupancy to assess conformational stability and flexibility. Furthermore, the detailed results are as follows-

**3.6.1  Root mean square deviations.**  The Root Mean Square Deviation (RMSD) analysis was conducted to evaluate the structural stability, conformational adaptability, and equilibrium dynamics of Mitoxantrone within the active sites of its four target proteins — Chk1 kinase (PDB ID: 2YEX), MDM2 E3 ubiquitin ligase (PDB ID: 4HG7), mTOR kinase domain (PDB ID: 4JSX), and PARP-1 DNA repair enzyme (PDB ID: 5DS3) — over a 100 ns MD trajectory. RMSD values were computed for the Cα atoms, protein backbone, and ligand heavy atoms across 1000 trajectory frames (100 ps per frame), enabling precise monitoring of global and local fluctuations during simulation. Across all complexes, the RMSD trajectories stabilised within the initial 10–15 ns, maintaining average deviations below 2.0 Å, indicative of thermodynamically stable binding with minimal large-scale structural perturbations. This behaviour confirms that Mitoxantrone forms robust and persistent interactions with its targets, consistent with the strong noncovalent binding predicted in docking and WaterMap analyses. In the Chk1–Mitoxantrone complex, the protein exhibited an initial RMSD of 1.14 Å and the ligand of 1.23 Å at 0.10 ns, reflecting rapid equilibration during the early phase. The RMSD remained consistent thereafter, with minor oscillations corresponding to loop flexibility around the ATP-binding cleft. At 100 ns, the protein RMSD reached 2.55 Å, while the ligand RMSD stabilised at 0.91 Å (Fig 7), signifying minimal ligand displacement from its initial binding orientation. Mitoxantrone maintained a stable, well-aligned conformation within the Chk1 catalytic pocket throughout the simulation. For the MDM2–Mitoxantrone complex, the protein displayed an RMSD of 0.70 Å, and the ligand 1.59 Å at 0.10 ns. The system reached equilibrium rapidly, with RMSD fluctuations remaining narrow throughout the trajectory. At 100 ns, the protein RMSD was 1.64 Å, and the ligand RMSD was 1.75 Å, reflecting a well-preserved binding geometry with minor conformational rearrangements of surface-exposed loops (Fig 7). The RMSD trends confirm that Mitoxantrone achieved a dynamically stable configuration within the hydrophobic binding cleft of MDM2, suggesting reliable ligand accommodation without significant conformational drift. The mTOR–Mitoxantrone complex displayed slightly higher fluctuations due to the protein's extensive and flexible kinase domain. At 1.30 ns, the protein RMSD reached 3.38 Å, and the ligand RMSD was 1.00 Å, indicating early adaptive conformational relaxation. By the end of the simulation (100 ns), the protein RMSD attained 5.63 Å, and the ligand RMSD was 1.49 Å (Fig 7). The observed higher protein deviation is attributed to global domain motions typical of the mTOR catalytic site, whereas the ligand RMSD remained stable, confirming tight anchoring of Mitoxantrone within the binding cavity despite backbone flexibility. This behaviour suggests that ligand binding does not disrupt the global fold but allows adaptive movement compatible with allosteric flexibility. In the PARP-1–Mitoxantrone complex, the protein and ligand RMSDs were 1.10 Å and 1.15 Å, respectively, at 0.10 ns, showing rapid convergence. Both maintained low fluctuations throughout the 100 ns trajectory, reflecting a highly stable binding interaction. At 100 ns,

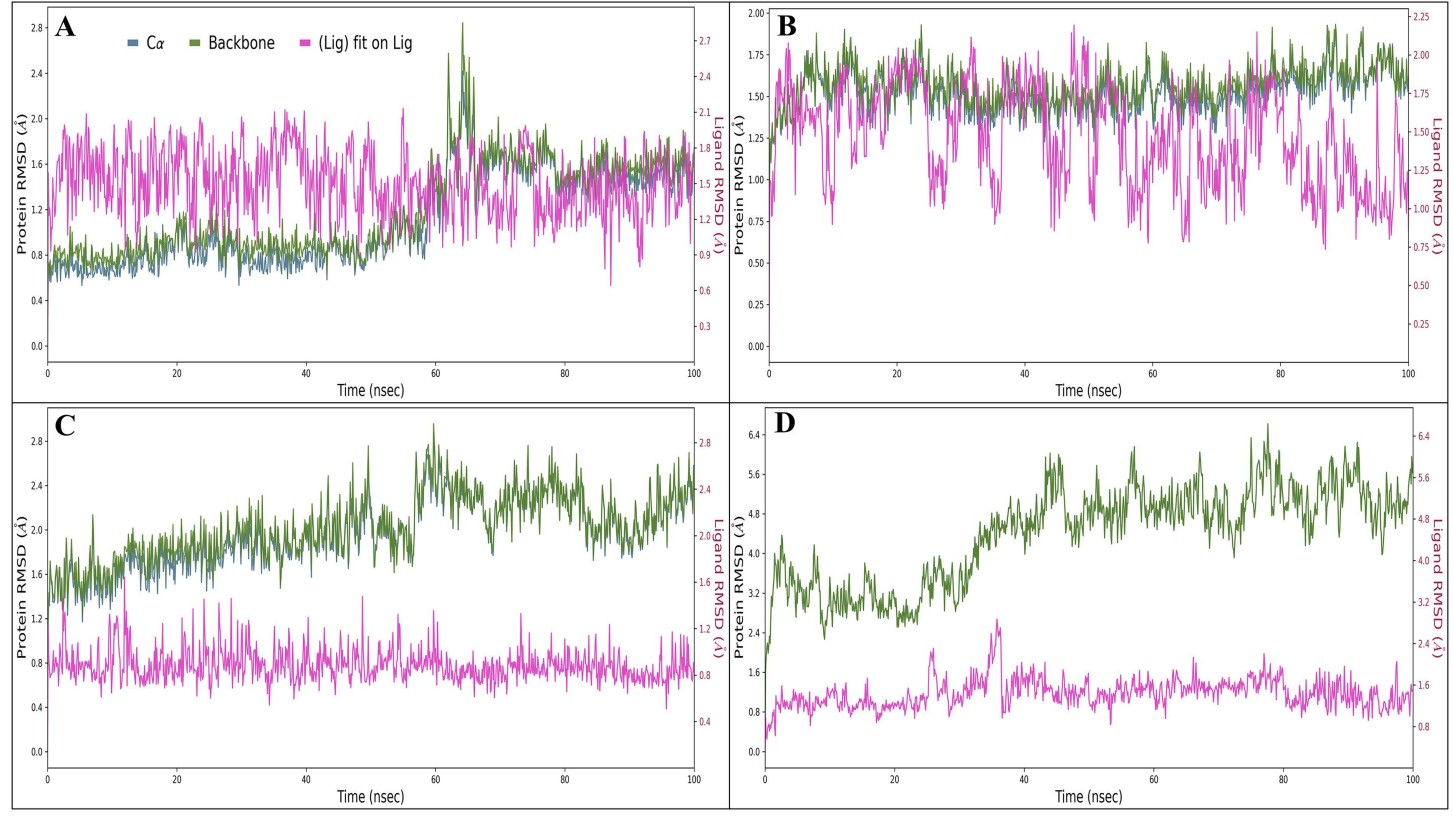

**Fig 7. The figure shows the Root Mean Square Deviation (RMSD) for Mitoxantrone in complex with A) Checkpoint kinase 1 (Chk1, PDB ID: 2YEX), B) MDM2 (Mouse double minute 2 homolog, PDB ID: 4HG7), C) mTOR (Mechanistic Target of Rapamycin kinase domain, PDB ID: 4JSX), and D) PARP-1 (Poly [ADP-ribose] polymerase 1 catalytic domain, PDB ID: 5DS3).**

the protein RMSD was 1.56 Å, and the ligand RMSD was 1.24 Å (Fig 7), consistent with a compact and rigid active-site conformation. The small deviation magnitudes suggest that the binding of Mitoxantrone to PARP-1 is conformationally stable and energetically favourable, with minimal structural rearrangement of the catalytic residues. Among all four systems, the 2YEX and 5DS3 complexes demonstrated the highest structural stability (RMSD ≤ 2.5 Å), indicating strong conformational resilience of the binding site and stable ligand retention. The 4HG7 complex exhibited slightly higher ligand mobility but remained within acceptable limits for a stable interaction. Conversely, the 4JSX complex showed the largest protein backbone fluctuations, reflecting the inherent flexibility of the mTOR kinase domain rather than instability in binding. The RMSD analysis confirms that Mitoxantrone maintains conformationally stable interactions across all four targets, supported by minimal backbone drift and consistent ligand RMSD trends. The results reinforce the thermodynamic and structural stability of the complexes, corroborating findings from docking, MIF, and WaterMap analyses.

**3.6.2 Root mean square fluctuations.** Root Mean Square Fluctuation (RMSF) analysis was performed to quantify the flexibility of individual amino acid residues in the protein–ligand complexes and to identify regions contributing to conformational stability or dynamic adaptability during the 100 ns molecular dynamics simulations. The RMSF profiles of Cα atoms revealed that most residues in the complexes maintained fluctuations below 2.0 Å, indicating structural rigidity and stable binding interactions between Mitoxantrone and the respective protein targets. Residues exhibiting higher fluctuations (>2.0 Å) were predominantly localised in loop regions or solvent-exposed domains, which are inherently flexible and less constrained by ligand interactions. In the Mitoxantrone–Checkpoint kinase 1 (Chk1, PDB ID: 2YEX) complex,

several residues—ALA2, VAL3, PRO4, PHE5, VAL6, GLU7, ARG29, VAL30, THR31, GLU32, VAL46, ASP47, CYS48, PRO49, and LYS270—displayed fluctuations exceeding 2.0 Å, corresponding primarily to terminal and loop regions. In contrast, multiple residues forming key hydrogen bonds and hydrophobic contacts, including GLN13, THR14, LEU15, GLY16, GLU17, GLY18, VAL23, ALA36, LYS38, VAL68, LEU84, GLU85, TYR86, CYS87, SER88, GLY89, GLU91, ASP94, GLU134, ASN135, LEU137, GLU140, SER147, and ASP148, demonstrated reduced flexibility, suggesting their involvement in maintaining complex stability and contributing to persistent ligand anchoring within the binding pocket (Fig 8). In the Mitoxantrone–MDM2 (PDB ID: 4HG7) complex, residues GLN18, ILE19, PRO20, ALA21, and GLN71 exhibited elevated RMSF values (>2.0 Å), reflecting flexible loop segments. Conversely, residues such as ACE17, GLN18, ILE19, PRO20, ALA21, SER22, GLN24, GLU25, MET50, LYS51, GLU52, LEU54, PHE55, TYR56, LEU57, GLY58, GLN59, ILE61, MET62, ARG65, TYR67, ASP68, GLN71, GLN72, HIS73, PHE91, VAL93, LYS94, GLU95, HIS96, ILE99, and TYR100 remained structurally stable throughout the simulation. These residues formed a network of hydrogen bonding and hydrophobic contacts stabilising Mitoxantrone within the binding cleft of MDM2 (Fig 8). The Mitoxantrone–mTOR (PDB ID: 4JSX) complex exhibited comparatively higher flexibility across several regions, with residues GLU1385, LYS1867, LYS1868, VAL1869, THR1870, GLY2040, GLU2041, ARG2042, ASN2043, VAL2044, LYS2090, SER2091, GLY2092, ASN2093, VAL2094, LYS2095, ASP2096, and VAL324 showing RMSF values above 2.0 Å, consistent with their positioning in

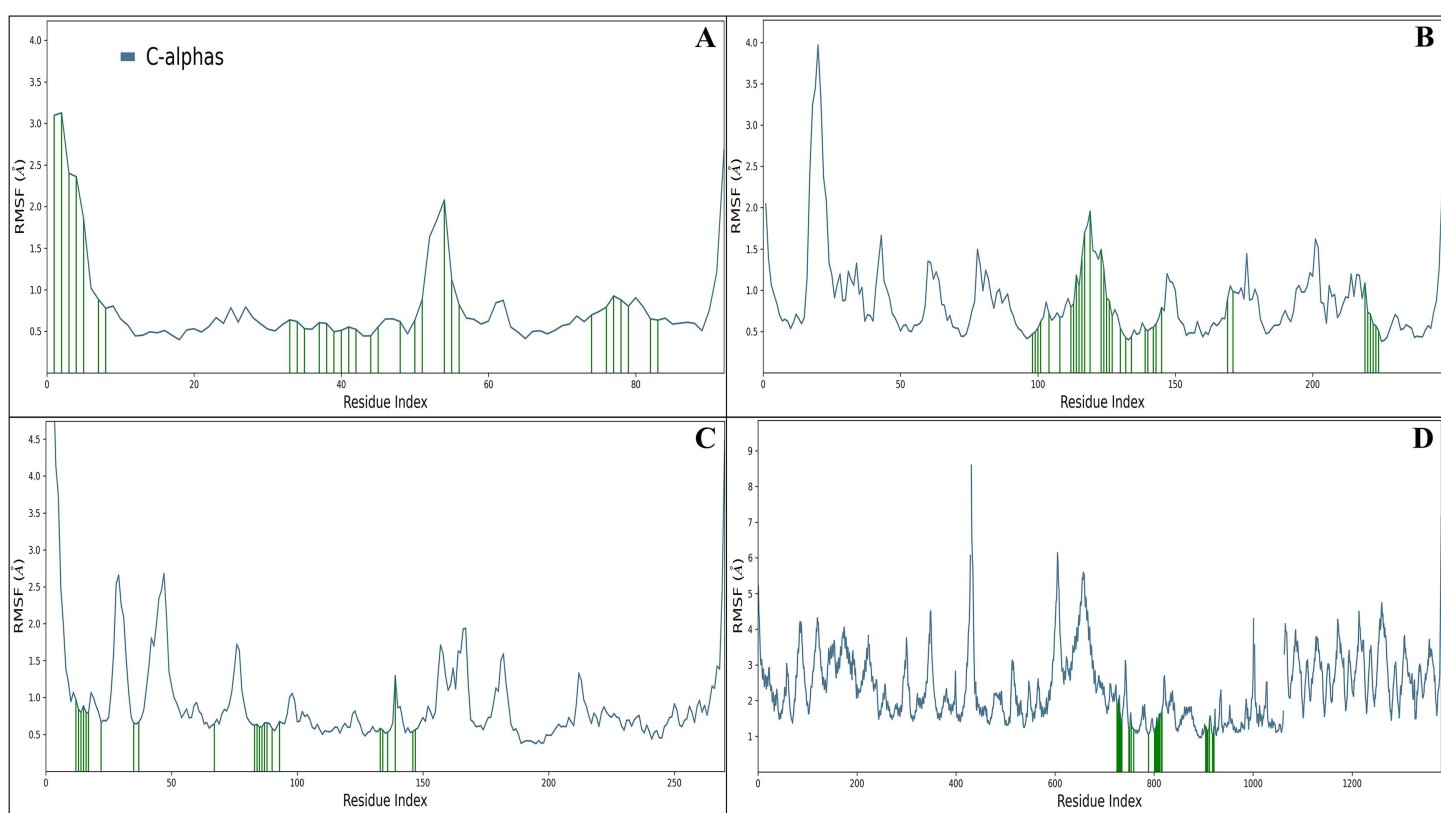

**Fig 8. The figure shows the Root Mean Square Fluctuation (RMSF) for the studied targets in complex with Mitoxantrone, with a legend indicating fluctuations in protein residues and the green line representing ligand interactions: A) Checkpoint kinase 1 (Chk1, PDB ID: 2YEX), B) MDM2 (Mouse double minute 2 homolog, PDB ID: 4HG7), C) mTOR (Mechanistic Target of Rapamycin kinase domain, PDB ID: 4JSX), and D) PARP-1 (Poly [ADP-ribose] polymerase 1 catalytic domain, PDB ID: 5DS3).**

flexible loop or terminal domains. Nevertheless, residues GLN2161, ILE2163, THR2164, SER2165, GLN2167, PRO2169, LYS2171, LEU2185, LYS2187, GLU2190, ASP2195,

TYR2225, ILE2237, GLY2238, TRP2239, VAL2240, HIS2242, CYS2243, ASP2244, THR2245, HIS2247, ALA2248, ARG2251, ASP2252, HIS2340, SER2342, ASN2343, MET2345, ARG2348, LEU2354, ILE2356, and ASP2357 formed persistent stabilising interactions that contributed to complex rigidity, indicating effective binding of Mitoxantrone in the mTOR active site (Fig 8). For the Mitoxantrone–PARP-1 (PDB ID: 5DS3) complex, residues SER765, SER781, GLY782, SER783, GLY784, SER785, GLY786, and GLY787 displayed higher fluctuations beyond 2.0 Å, mainly corresponding to flexible loop regions distal to the ligand-binding site. In contrast, residues HIS862, GLY863, SER864, ARG865, ASN868, ILE872, GLY876, LEU877, ARG878, ILE879, ALA880, PRO881, GLU883, THR887, GLY888, TYR889, MET890, PHE891, GLY894, TYR896, ALA898, LYS903, SER904, ASN906, TYR907, HIS909, LYS933, ALA935, SER983, LEU984, LEU985, TYR986, ASN987, and GLU988 exhibited low RMSF values, confirming their role in maintaining a compact and well-stabilised binding environment for Mitoxantrone (Fig 8). The RMSF profiles across all four target complexes confirm that Mitoxantrone binding induces localised flexibility in peripheral or loop regions, while preserving the structural integrity of the catalytic or ligand-binding domains—suggesting that Mitoxantrone promotes an optimal balance between conformational adaptability and complex stability throughout the MD Simulations' trajectory (Fig 8).

**3.6.3 Analysis of simulation interaction.** Simulation Interaction Diagram (SID) analysis was conducted to elucidate the nature and persistence of noncovalent interactions between Mitoxantrone and its protein targets, including hydrogen bonding, electrostatic (salt-bridge) interactions, hydrophobic contacts, and aromatic π–π stacking. These interactions collectively define the ligand-protein complexes' stability, specificity, and binding affinity throughout the molecular dynamics (MD) simulations. In the Mitoxantrone–Checkpoint kinase 1 (Chk1, PDB ID: 2YEX) complex, Mitoxantrone established a dense hydrogen-bonding network involving key residues GLU17, SER88, ASP94, TYR86, GLN13, and GLY18, as well as GLU134, which interacted with water-mediated bridges via two $N^+H_2$ donor atoms of the ligand. Additional hydrogen bonds were formed between the ligand's NH groups and residues GLU17 and CYS87, while the carbonyl O atom of Mitoxantrone engaged CYS87 directly. Furthermore, four hydroxyl (-OH) ligand groups participated in hydrogen bonding with GLU85, CYS87, ASP148, and SER147, as well as with LYS38 and ASP94 through water bridges. A notable electrostatic salt bridge was detected between the ASP148 residue and Mitoxantrone's protonated amine ($N^+H_2$), contributing significantly to electrostatic stabilisation of the complex (Fig 9). The Mitoxantrone–MDM2 (PDB ID: 4HG7) complex exhibited extensive hydrogen bonding and π-interactions, further reinforcing its structural stability. Hydrogen bonds were formed with GLN24, HIS96, and TYR67 residues, while water-mediated hydrogen bonds involved LYS51 and GLY58 via the ligand's $N^+H_2$ groups. The HIS96 and TYR67 residues additionally formed hydrogen bonds through NH donors, and the ligand's O atoms interacted with TYR67 and HIS96 via bridging water molecules. Furthermore, hydroxyl groups of Mitoxantrone interacted with TYR67, GLN72, GLN24, TYR100, GLY58, and GLU25. Three π–cation interactions were observed with PHE55, TYR67, and HIS96 through the ligand's amine groups, suggesting a significant contribution of aromatic electrostatics to the overall stabilisation of the MDM2–Mitoxantrone complex (Fig 9). Multiple stabilising interactions were detected for the Mitoxantrone–mTOR (PDB ID: 4JSX) complex. Hydrogen bonds were primarily formed with residues ASN2343, ASP2357, SER2342, and CYS2243, whereas THR2245, ASP2244, TRP2239, and VAL2240 interacted through water bridges involving the ligand's $N^+H_2$ atoms. The ligand's NH donor formed additional direct contacts with CYS2243, and its carbonyl O atom interacted with VAL2240 and CYS2243. Hydroxyl groups of Mitoxantrone established hydrogen bonds with ASP2244, VAL2240, GLU2190, and ASP2357. The complex was further stabilised by π–π stacking between TYR2225 and TRP2239 through two aromatic rings, and salt bridges were observed between the negatively charged residues GLU2190 and ASP2244 and the protonated $N^+H_2$ groups of Mitoxantrone, reinforcing both electrostatic and aromatic complementarity at the binding interface (Fig 9). The SID analysis revealed strong polar and aromatic stabilisation patterns in the Mitoxantrone–PARP-1 (PDB ID: 5DS3) complex. Hydrogen bonds were established between the ligand and residues GLY888, GLU988, TYR889, TYR907,

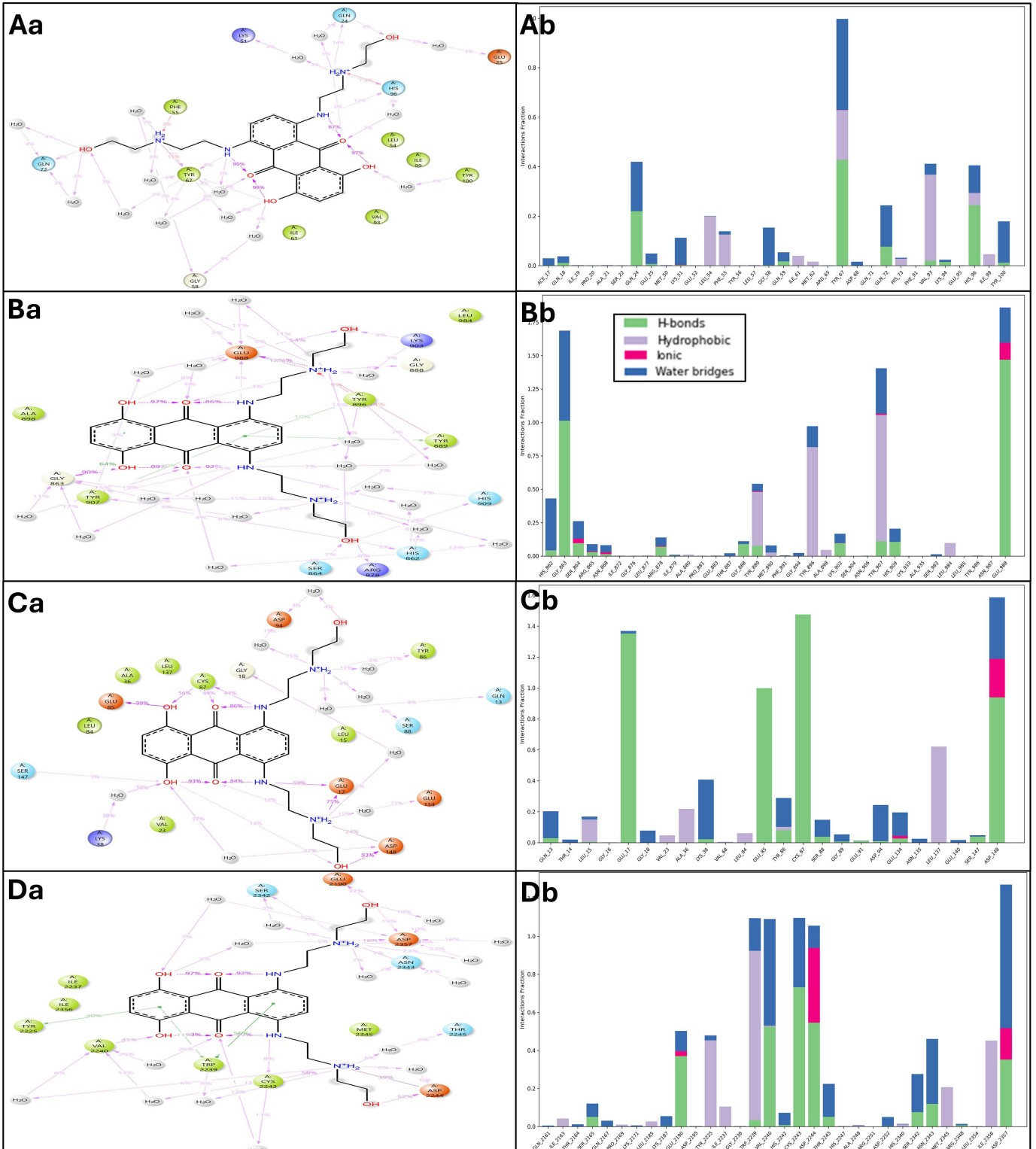

**Fig 9. The figure shows the Simulation Interaction Diagram (SID) and its histogram representation for the studied targets in complex with Mitoxantrone with Aa, Ab) Checkpoint kinase 1 (Chk1, PDB ID: 2YEX), Ba, Bb) MDM2 (Mouse double minute 2 homolog, PDB ID: 4HG7), Ca, Cb) mTOR (Mechanistic Target of Rapamycin kinase domain, PDB ID: 4JSX), and Da, Db) PARP-1 (Poly [ADP-ribose] polymerase 1 catalytic domain, PDB ID: 5DS3).**

GLY863, and HIS862, with water-mediated contacts involving the ligand's $N^+H_2$ groups. Additional NH donor interactions were observed with GLU988 and HIS862, while the O atoms of the ligand participated in hydrogen bonding with GLU988, GLY863, and TYR896. Hydroxyl functionalities of Mitoxantrone further stabilised the complex through interactions with SER864, HIS909, ARG878, TYR907, GLY863, GLU988, LYS903, and TYR896. Two π–cation interactions were formed with TYR896 and TYR889 through the $N^+H_2$ group of the ligand, and four π–π stacking interactions were detected involving TYR896, TYR889, and TYR907 across both aromatic rings of Mitoxantrone. A distinct salt bridge was also observed between the negatively charged GLU988 residue and the protonated $N^+H_2$ moiety of the ligand, indicating strong electrostatic complementarity and contributing to the exceptional stability of the PARP-1–Mitoxantrone complex (Fig 9). The SID analysis across all four complexes revealed that Mitoxantrone forms a consistent combination of direct and water-mediated hydrogen bonds, complemented by electrostatic and aromatic interactions. These cooperative forces underpin the thermodynamic stability and conformational rigidity observed in the RMSD and RMSF analyses, suggesting robust ligand accommodation and high-affinity binding across multiple oncogenic targets.

### 3.7 Analysis of binding free energy and other profiles

To quantify the binding affinities and energetic stabilities of Mitoxantrone within its protein complexes, MM-GBSA calculations were performed using 100 ns MD simulation trajectories for each target—Checkpoint kinase 1 (Chk1, PDB ID: 2YEX), Mouse double minute 2 homolog (MDM2, PDB ID: 4HG7), Poly(ADP-ribose) polymerase 1 (PARP-1, PDB ID: 5DS3), and Mechanistic target of rapamycin kinase domain (mTOR, PDB ID: 4JSX). The MM-GBSA method decomposes the total binding free energy (ΔG_bind) into contributions from molecular mechanics energy terms (van der Waals, electrostatics) and solvation free energies (polar and nonpolar components), thereby providing a physically grounded measure of the thermodynamic favourability of ligand binding under physiological-like conditions. The average binding free energy (ΔG_bind) values obtained for the four complexes were as follows: 2YEX (−80.44 kcal/mol), 4HG7 (−50.21 kcal/mol), 5DS3 (−85.21 kcal/mol), and 4JSX (−50.21 kcal/mol). These results indicate that the complexes with PARP-1 (5DS3) and Chk1 (2YEX) are energetically more favourable, reflecting stronger and more stable ligand–protein interactions relative to MDM2 and mTOR, which exhibit comparatively weaker binding affinities. The standard deviations (SD) of ΔG values ranged between 5.3 and 7.25 kcal/mol, signifying minimal energetic fluctuation throughout the simulation, implying the ligand's structural convergence and dynamic stability within the binding pocket. The narrow ΔG further corroborates these stability ranges for 2YEX (−97.89 to −65.64 kcal/mol) and 5DS3 (−104.77 to −60.85 kcal/mol), whereas wider distributions in 4HG7 and 4JSX (approximately −72.36 to −25.67 kcal/mol) suggest transient conformational transitions and occasional water-mediated rearrangements within the binding cavity. The non-solvated free energy (ΔG_NS) represents the vacuum molecular mechanics contribution (sum of electrostatic and van der Waals energies) before inclusion of solvent effects. The ΔG_NS values were observed as −85.86 kcal/mol (2YEX), −54.66 kcal/mol (4HG7), −90.64 kcal/mol (5DS3), and −54.66 kcal/mol (4JSX). These values emphasise that hydrophobic and electrostatic complementarity primarily drive Mitoxantrone's binding affinity in 2YEX and 5DS3, with the deeper and more enclosed pockets in these targets enabling stronger non-solvated interactions. The higher (less negative) ΔG_NS in MDM2 and mTOR suggests that solvent-exposed binding sites reduce the contribution of nonpolar contacts, resulting in weaker net binding. The standard deviation of the non-solvated component (ΔG_NS SD: 5.3–7.25 kcal/mol) closely parallels that of total ΔG, indicating consistent electrostatic–hydrophobic compensation across the simulation frames. This balance suggests that even during local conformational fluctuations, the essential binding geometry of Mitoxantrone remained conserved. The trajectory-based analysis (1001 frames) further substantiates that the binding energies remained persistently negative across the entire 100 ns simulation (Fig 10, SS08), reaffirming the thermodynamic stability of the complexes under dynamic solvent and temperature conditions. Among the complexes, Mitoxantrone–PARP-1 (5DS3) exhibited the most favourable binding free energy (−85.21 kcal/mol), signifying strong electrostatic and hydrogen-bonding contributions, consistent with multiple stabilising interactions observed in the SID analysis (e.g., TYR896, GLU988, and HIS862). The Chk1 complex (2YEX) also displayed

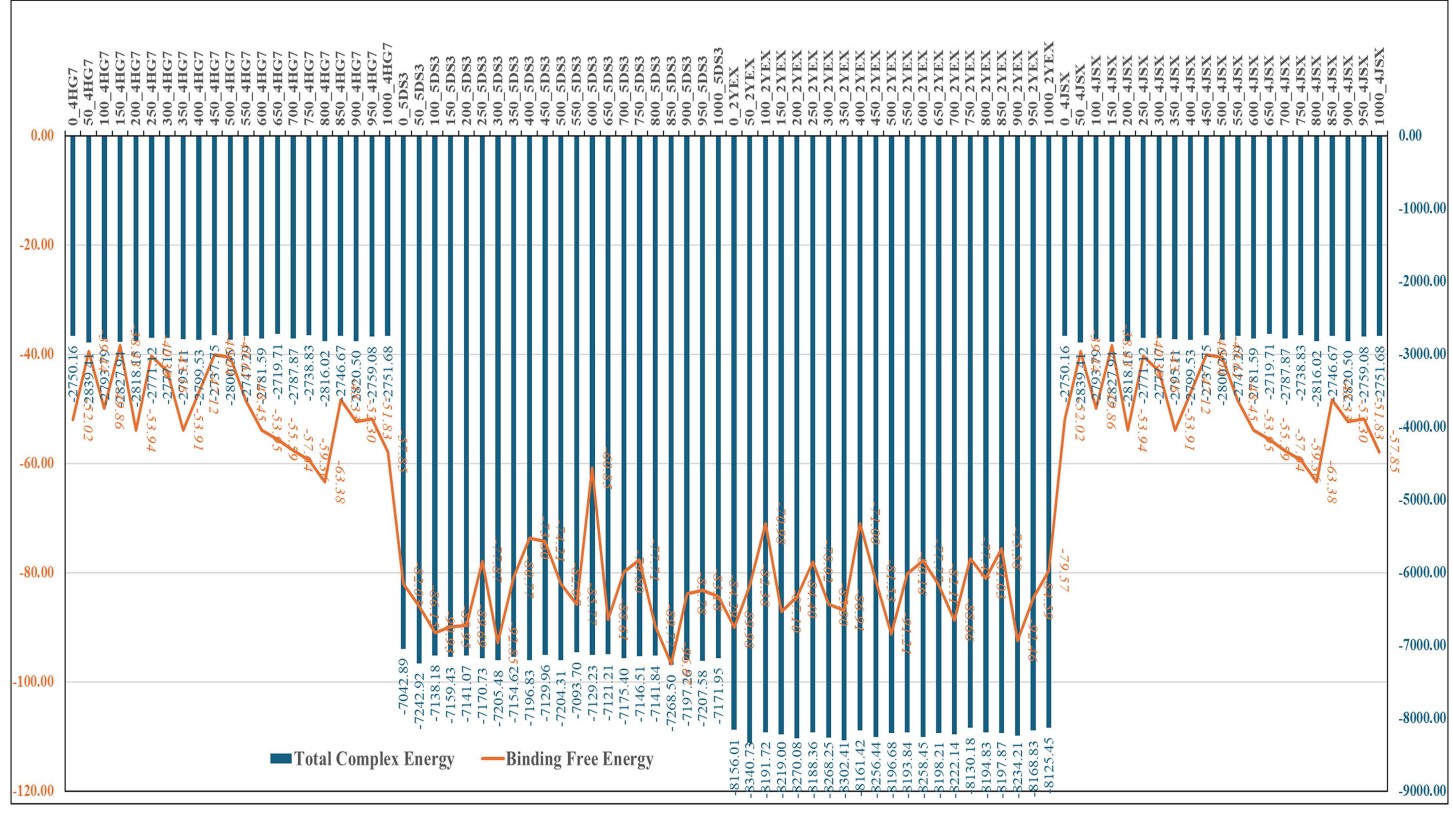

**Fig 10. The figure shows the Binding Free and Total Complex Energy of Mitoxantrone in complex with Checkpoint kinase 1 (Chk1, PDB ID: 2YEX), MDM2 (Mouse double minute 2 homolog, PDB ID: 4HG7), mTOR (Mechanistic Target of Rapamycin kinase domain, PDB ID: 4JSX), and PARP-1 (Poly [ADP-ribose] polymerase 1 catalytic domain, PDB ID: 5DS3), computed on 100 ns MD Simulation Trajectories.**

high binding affinity (−80.44 kcal/mol), driven by persistent hydrogen bonding and π–π stacking interactions involving TYR86, GLU17, and ASP148. In contrast, MDM2 (4HG7) and mTOR (4JSX) showed comparatively weaker binding energies (−50.21 kcal/mol each), corresponding to shallower or more solvent-accessible binding pockets where water competition reduces net interaction strength. ΔG_bind (kcal/mol): Represents the net thermodynamic favourability of ligand binding; lower (more negative) values indicate stronger affinity and higher structural stability. The MM-GBSA calculations confirm that Mitoxantrone forms energetically stable and thermodynamically favourable complexes, particularly with Chk1 (2YEX) and PARP-1 (5DS3), consistent with their lower RMSD and RMSF values. In conjunction with SID and WaterMap analyses, these data underscore that enhanced stability arises from an optimal interplay of electrostatic, hydrophobic, and aromatic interactions. The consistent correlation between the total complex energy and binding free energy profiles (Fig 10, SS07) validates the reliability of the computed energetics and supports Mitoxantrone's potential as a multitarget anticancer agent.

## 4 Discussion

The present integrated computational investigation elucidates the potential of Mitoxantrone as a multitargeted therapeutic candidate for ageing-associated cancers, grounded on an exhaustive suite of structural, electronic, docking, pharmacokinetic, hydration-thermodynamic, and dynamics analyses. By merging the protein-preparation energy landscapes, molecular docking comparisons (with control Doxorubicin), DFT profiling, MIFs analysis, QikProp-based pharmacokinetic

predictions, WaterMap hydration-site mapping, MD stability metrics (RMSD and RMSF) and MM-GBSA free energy snapshots, we provide a mechanistically coherent and quantitative argument for Mitoxantrone's repositioning in the context of cancers driven by ageing-related biological processes. From the outset, the protein preparation energies for the four targets (Chk1, MDM2, mTOR, PARP-1) revealed highly negative total and potential energies (e.g., –351.5 to –5 231.6 kcal/mol), indicating that the receptor conformations were in well-relaxed minima and thereby suitable for subsequent docking and dynamic investigations. High magnitudes of non-bonded (Lennard–Jones and electrostatic) contributions underscore the importance of van der Waals and Coulombic packing in these macromolecular systems. Establishing stable receptor models is fundamental when exploring multitarget ligand binding; our workflow thus ensures that subsequent docking metrics reflect ligand complementarity rather than artefacts of poorly minimised structures. The docking results strengthen Mitoxantrone's case: across all four targets, docking scores ranged from –6.23 to –16.04 kcal/mol, and MM-GBSA ΔG_bind values ranged between –49.19 and –85.14 kcal/mol. Notably, the best affinities were observed for Chk1 (–16.04 kcal/mol; ΔG_bind –85.14 kcal/mol) and PARP-1 (–10.23 kcal/mol; ΔG_bind –78.27 kcal/mol), which are both strongly implicated in DNA damage response and checkpoint regulation in ageing tissues. When compared to Doxorubicin, which exhibited docking scores of –5.21 to –8.05 kcal/mol and ΔG_bind of –39.99 to –66.04 kcal/mol, Mitoxantrone consistently demonstrated superior predicted binding affinity and ligand efficiency (–2.661 to –1.537 kcal/mol/atom versus –1.693 to –1.025 for the control). This comparative advantage suggests that Mitoxantrone may engage target pockets more efficiently and potentially with broader target coverage. Mechanistically, the docking and subsequent interaction fingerprint analyses reveal that Mitoxantrone engages an ensemble of key residues via hydrogen bonds, π–π stacking, π–cation interactions, and salt bridges across different target proteins. For instance, in the Chk1 complex, hydrogen bonds with SER88, GLU134 and CYS87 and a salt-bridge with ASP148 anchor the ligand within the ATP-binding cleft. In the mTOR complex, π–π interactions with TYR2225 and TRP2239 plus salt bridges with GLU2190 and ASP2244 highlight engagement with regulatory motifs. The breadth of interacting residues (e.g., GLU, ASP, TYR, HIS, ARG) across all targets suggests that Mitoxantrone possesses structural flexibility and functional specificity, enabling its activity across multiple cancer-relevant nodes.

The DFT results of Mitoxantrone further substantiate its suitability as a multitarget agent in ageing-associated cancer contexts. The HOMO (–0.3857 eV) and LUMO (–0.3115 eV) energies yield a narrow gap (~0.074 eV), indicative of a highly polarisable molecular entity with the capability for charge transfer and electronic adaptability in dynamic protein environments. The average local ionisation energy (ALIE) values, with minima around ~318.98 kcal/mol and maxima ~524.94 kcal/mol, reveal zones of both electron-rich (susceptible to electrophilic attack) and electron-stable regions, which may facilitate cooperative binding through hydrogen-bond donor or acceptor roles. The electrostatic potential (ESP) surface statistics (mean ~123.9 kcal/mol, max ~183.18, min ~68.68) show a well-defined polar surface architecture, compatible with engaging both hydrophobic pockets (through π systems) and polar/charged enclaves (through protonated amines and hydroxyl groups). In the ageing-cancer paradigm, wherein senescence, DNA damage accumulation and altered redox signalling play pivotal roles, a ligand that combines strong binding with electronic adaptability provides a distinct mechanistic advantage. Pharmacokinetic profiling via QikProp offers further insight into Mitoxantrone's potential translational value. The predicted human oral absorption for Mitoxantrone was ~19.2% (versus 0% for Doxorubicin), with a balanced QPlogPo/w (0.366) and higher solubility (QPlogS –0.433), contrasted with Doxorubicin's –0.453 and –2.454, respectively. Although neither compound is ideal for oral delivery, the comparative advantage suggests Mitoxantrone may have better systemic availability. Other parameters, such as moderate plasma protein binding (QPlogKhsa –0.384) and a cardiac channel inhibition risk (QPlogHERG –6.395) comparable to Doxorubicin, emphasise that safety considerations remain, but relative drug-likeness and improved rule-of-five compliance (Mitoxantrone violated only one rule vs three for the control) further support repurposing potential. For comparison, biologically active anticancer molecules such as Doxorubicin, Daunorubicin, Imatinib, and other quinone- or heteroaromatic-based drugs typically exhibit HOMO–LUMO gaps in the range of 2–5 eV, as reported in previous DFT studies. The markedly lower gap of Mitoxantrone (ΔE = 0.074 eV), therefore,

indicates an exceptionally high degree of electronic polarizability and charge-transfer capability, which is consistent with its strong docking interactions and multitarget binding behaviour. In the context of ageing-associated cancers, where polypharmacology and target network modulation are increasingly recognised, having a molecule with favourable ADME descriptors and a broader target spectrum is advantageous. Hydration thermodynamics via WaterMap analysis bolsters the structural insights: across all four complexes, the displacement of high-energy water clusters (e.g., near GLU, ASP, TYR residues) was a recurrent theme. For instance, in the 4HG7 complex, the identification of displaced hydration sites adjacent to ASP3 and GLU5 indicates favourable entropy gain upon binding; in the 2YEX complex, retention of key hydration sites near ASP3 and GLU5 alongside ligand interactions suggests enthalpy–entropy compensation and enhanced binding specificity. The ability of Mitoxantrone to exploit hydrophobic patches (VAL6, LEU5, ALA4) and polar nodes (GLU, ASP) across distinct binding sites reflects its structural versatility—particularly relevant in ageing cancers where microenvironmental changes (e.g., senescent-associated secretory phenotype, extracellular matrix remodelling) alter hydration and solvent dynamics. The hydration mapping thus supports the ligand's ability to adapt and stabilise in heterogeneous tumour-microenvironment conditions.

MD simulation results (RMSD and RMSF analysis) validate the structural and dynamic robustness of the ligand-protein complexes. RMSD trajectories remained within acceptable bounds (mostly <2 Å) for complexes with Chk1 (protein ~2.55 Å; ligand ~0.91 Å at 100 ns) and PARP-1 (protein ~1.56 Å; ligand ~1.24 Å at 100 ns), indicating tight binding and minimal conformational drift. Even for the mTOR complex, where protein RMSD reached ~5.63 Å owing to inherent domain flexibility, the ligand RMSD remained low (~1.49 Å), signifying a persistent anchor within a dynamic pocket. RMSF profiles further reveal that residues exhibiting higher flexibility (>2 Å) were predominantly loop or terminal regions (e.g., ALA2, VAL3, GLU7 in 2YEX; GLU1385, VAL324 in 4JSX), whereas residues forming stable intermolecular contacts (e.g., THR14, LEU15 in 2YEX; SER2165, GLN2167 in 4JSX) remained tightly constrained. Such dynamic signatures imply that Mitoxantrone binding enforces rigidity in functionally critical regions (active site, regulatory loop) while permitting peripheral flexibility—an important attribute in targeting ageing-driven cancers where protein mutational landscapes and allosteric conformations may diverge from canonical forms. The MM-GBSA analysis delivers thermodynamic quantification of binding efficacy across the four complexes. The average ΔG_bind values (–80.44 kcal/mol for 2YEX; –50.21 kcal/mol for 4HG7; –85.21 kcal/mol for 5DS3; –50.21 kcal/mol for 4JSX) confirm that Chk1 and PARP-1 complexes are energetically most favourable. The non-solvated energies (ΔG_NS) follow a parallel trend (–85.86 and –90.64 kcal/mol, respectively), emphasising strong van der Waals and Coulombic complementarity in those systems. Fluctuation ranges and standard deviations (SD～5–7 kcal/mol) illustrate that binding remained stable over the 100 ns trajectories. Mitoxantrone does not merely dock well in practice, but sustains energetically favourable binding under dynamic, solvated conditions—a critical threshold for effective *in-vivo* translation. The weaker energies recorded for MDM2 and mTOR suggest that binding in more solvent-exposed or conformationally plastic pockets is less favourable, correlating with docking and dynamic data. In the framework of ageing-associated cancers—where target promiscuity, DNA damage accumulation (via PARP-1), checkpoint dysregulation (via Chk1), and nutrient-sensing networks (via mTOR) are central hallmarks—the ability of Mitoxantrone to bind strongly to two of these major nodes (Chk1 and PARP-1) with moderate affinity across others positions it as a plausible "hub-targeting" ligand for poly-modal cancer therapy.

The present data support the proposition that Mitoxantrone may function as a game-changer molecule in ageing-associated cancers through several convergent mechanisms. Firstly, strongly binding to key ageing-associated cancer targets (Chk1 and PARP-1) may effectively intercept dysregulated cellular processes in the aged and cancerous tissue, such as DNA damage response, senescence escape and proliferative checkpoint failure. The literature confirms that ageing and cancer share mechanistic hallmarks—telomere attrition, stem-cell exhaustion, oncogene activation, altered nutrient-sensing (e.g., mTOR signalling) and disrupted immune surveillance. By targeting Chk1 and PARP-1, Mitoxantrone may reinvigorate checkpoint fidelity and hinder the survival of senescence-associated pre-malignant clones. Secondly, its electronic and pharmacokinetic profile (narrow HOMO–LUMO gap, balanced solubility, moderate oral absorption)

suggests adaptability and systemic applicability, important for treating heterogeneous ageing-cancer phenotypes where multi-compartment dissemination is common. Thirdly, the hydration and dynamics analyses show that Mitoxantrone can stabilise binding in variable microenvironments—reflective of ageing tissue microenvironments (increased fibrosis, altered ECM, chronic inflammation). Fourthly, the comparative advantage over Doxorubicin (improved docking/efficiency, better pharmacokinetic profile, less predicted cardiac risk) provides a clear incentive for repositioning Mitoxantrone beyond its conventional use in haematological and metastatic cancers. Fourthly, evidence beyond computational modelling supports its modulatory potential: recent studies indicate that Mitoxantrone triggers immunogenic cell death in prostate cancer via a p53-dependent PERK pathway and modulates TET-mediated epigenetic pathways in leukaemia. These mechanisms dovetail with ageing-related epigenetic drift and immune dysfunction, further strengthening the rationale for its repurposing.

In the broader therapeutic landscape of ageing-associated malignancies, where interventions targeting cellular senescence, nutrient-sensing pathways, DNA repair mechanisms, and immunosenescence are gaining prominence, Mitoxantrone emerges as a uniquely positioned molecule. Its ability to integrate classical cytotoxic activity with checkpoint modulation and epigenetic and immunomodulatory effects provides a strategic advantage over conventional agents. By potentially synergising with senolytic or senomorphic therapies, Mitoxantrone could facilitate the selective clearance or functional reprogramming of damaged cellular populations, mitigate the senescence-associated secretory phenotype, and reduce the likelihood of secondary malignancies in ageing cancer survivors. Such multimodal actions could address tumour progression and the deleterious sequelae of ageing. Nonetheless, several caveats warrant consideration. The present computational findings, though robust, remain predictive and necessitate empirical validation in ageing-relevant cancer models and non-malignant tissue systems. Moreover, Mitoxantrone's established toxicity profile—particularly its dose-dependent cardiotoxicity and propensity to induce senescence in non-tumour cells—may constrain its clinical applicability in older populations. Consequently, future investigations should integrate *in-vitro* and *in-vivo* ageing-cancer models, evaluate dose–response relationships, and explore combination regimens incorporating senolytic or cardioprotective agents to refine their therapeutic window. The *in-silico* evidence presented here indicates that Mitoxantrone possesses mechanistic and pharmacokinetic attributes that justify its consideration as a repurposed candidate for ageing-associated cancers; however, translational and clinical validation remain essential prerequisites before therapeutic repositioning can be realised.

## 5 Limitations and future directions

Despite the comprehensive and integrative computational workflow employed in this study—spanning docking, DFT, MIFs, pharmacokinetics, hydration thermodynamics, MD simulations, and MM-GBSA free-energy calculations—several limitations must be acknowledged to contextualise the findings. First, the in-silico results, though internally consistent and mechanistically coherent, remain predictive in nature. The protein structures used represent static or semi-static conformations that may not fully capture the conformational heterogeneity, post-translational modifications, or age-associated structural perturbations present in biological systems. Similarly, the docking and MD simulations were performed under idealised solvent and thermodynamic conditions, which do not fully reproduce the complex biochemical environments of ageing tissues, including altered redox states, elevated inflammatory cytokines, matrix stiffness, and senescence-associated microenvironmental cues.

The pharmacokinetic predictions, derived from QikProp and limited descriptor sets, provide initial guidance but do not replace absorption, distribution, metabolism, excretion, and toxicity (ADMET) profiling in relevant biological models. Additionally, Mitoxantrone's known cardiotoxicity and potential to induce senescence in non-malignant cells pose significant translational challenges that computational methods alone cannot resolve. The hydration thermodynamics and MM-GBSA calculations, although valuable, rely on approximations regarding solvent behaviour, entropic contributions, and long-range electrostatics, which may underestimate or overestimate binding affinities in vivo. Moreover,

the multi-target evaluation focused on four major ageing-associated cancer nodes; while insightful, ageing-related cancers involve wider signalling crosstalk, and additional targets may further refine or complicate Mitoxantrone's polypharmacological profile.

In-vitro biochemical assays, including target-specific inhibition studies and kinetic analyses in senescent or pre-senescent cancer models, are essential to substantiate the predicted binding affinities and interaction fingerprints. In-vivo studies in ageing-relevant animal models will be critical to evaluate pharmacodynamics, biodistribution, toxicity, and therapeutic indices. Given Mitoxantrone's dual potential to induce cytotoxicity and exacerbate ageing phenotypes, future work should explore dose optimisation, structural analogues with reduced cardiotoxicity, and combination therapies incorporating senolytics, senomorphics, or cardioprotective agents. Expanding the computational framework to incorporate enhanced sampling techniques, QM/MM hybrid simulations, and machine-learning-based polypharmacology predictions would further refine mechanistic insights. Finally, exploring Mitoxantrone's interactions across broader signalling networks—such as epigenetic regulators, immune-senescence pathways, and nutrient-sensing systems—may reveal additional mechanistic nodes that strengthen its candidacy for ageing-associated cancer therapy.

## 6 Conclusion

The comprehensive computational investigation integrating molecular docking, DFT, pharmacokinetics, WaterMap, MD simulations, interaction fingerprinting, and MM-GBSA analyses collectively supports the repositioning of Mitoxantrone as a promising multitarget therapeutic candidate for ageing-associated cancers. The docking and free-energy profiles demonstrated strong and stable binding across major cancer-ageing targets—Checkpoint kinase 1 (Chk1, PDB 2YEX), MDM2 (PDB 4HG7), PARP-1 (PDB 5DS3), and mTOR (PDB 4JSX)—with docking scores ranging from −6.23 to −16.04 kcal/mol and MM-GBSA ΔG_bind values between −49.19 and −85.21 kcal/mol. The strongest binding was observed for Chk1 (docking score −16.04 kcal/mol; ΔG_bind −80.44 kcal/mol) and PARP-1 (docking score −10.23 kcal/mol; ΔG_bind −85.21 kcal/mol). These two proteins play major roles in DNA-damage signalling and checkpoint control in ageing and cancer cells. The MIFs showed that Mitoxantrone interacts with a wide network of key residues—mainly GLU, ASP, TYR, HIS, and ARG—which reflects strong electrostatic and structural complementarity within these binding pockets. DFT analysis confirmed a narrow HOMO–LUMO gap (~0.074 eV), indicating electronic adaptability crucial for dynamic protein interactions. Pharmacokinetic evaluation showed improved solubility (QPlogS = −0.433) and moderate absorption (19.2%) relative to Doxorubicin, with fewer Lipinski rule violations, supporting its drug-likeness. WaterMap analysis highlighted favourable displacement of high-energy hydration sites, further contributing to binding enthalpy and entropy. MD simulations confirmed stability across 100 ns trajectories, with RMSD fluctuations < 2.5 Å for most complexes and ligand RMSD ≤ 1.5 Å, signifying minimal conformational drift. MM-GBSA ΔG(NS) values (−85.86 to −90.64 kcal/mol) affirmed stable, energetically favourable binding throughout the simulation window. These multi-parameter results underscore Mitoxantrone's capability to simultaneously engage and modulate multiple ageing-associated cancer targets. Its robust thermodynamic stability, favourable electronic reactivity, and improved pharmacokinetic indices position it as a potential game-changer molecule in managing cancers driven by ageing hallmarks, particularly those involving genomic instability and deregulated checkpoint signalling. Nonetheless, further *in-vitro* and *in-vivo* validations are essential to corroborate these *in-silico* predictions and delineate their therapeutic safety margins in ageing populations.

## Supporting information

**S1 File. Ageing associated cancer DataFiles.**
(XLSX)

## Acknowledgments

This research work was funded by Institutional Fund Projects under grant number IFPNC: 009-117-2020. The authors gratefully acknowledge technical and financial support from the Ministry of Education and King Abdulaziz University, Jeddah, Saudi Arabia. The funders had no role in study design, data collection and analysis, decision to publish, or preparation of the manuscript.

## Author contributions

**Conceptualization:** Mohammed H. Al-Qahtani, Mourad Assidi, Abdelbaset Buhmeida, Asma Almuhammadi, Nofe Ateq Alganmi.

**Data curation:** Mohammed H. Al-Qahtani, Asma Almuhammadi, Peter Natesan Pushparaj, Nofe Ateq Alganmi.

**Formal analysis:** Mohammed H. Al-Qahtani, Mourad Assidi, Asma Almuhammadi, Nofe Ateq Alganmi.

**Funding acquisition:** Mohammed H. Al-Qahtani.

**Investigation:** Mohammed H. Al-Qahtani, Abdelbaset Buhmeida, Nofe Ateq Alganmi.

**Methodology:** Mohammed H. Al-Qahtani, Mourad Assidi, Asma Almuhammadi, Peter Natesan Pushparaj.

**Project administration:** Abdelbaset Buhmeida, Asma Almuhammadi, Nofe Ateq Alganmi.

**Resources:** Mohammed H. Al-Qahtani, Mourad Assidi, Asma Almuhammadi, Peter Natesan Pushparaj, Nofe Ateq Alganmi.

**Software:** Mourad Assidi, Abdelbaset Buhmeida, Nofe Ateq Alganmi.

**Supervision:** Mohammed H. Al-Qahtani, Abdelbaset Buhmeida, Asma Almuhammadi, Peter Natesan Pushparaj.

**Validation:** Mohammed H. Al-Qahtani, Mourad Assidi, Abdelbaset Buhmeida.

**Visualization:** Mourad Assidi, Asma Almuhammadi, Nofe Ateq Alganmi.

**Writing – original draft:** Mohammed H. Al-Qahtani, Mourad Assidi, Abdelbaset Buhmeida, Peter Natesan Pushparaj.

**Writing – review & editing:** Mohammed H. Al-Qahtani, Asma Almuhammadi, Peter Natesan Pushparaj, Nofe Ateq Alganmi.

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
