## [Decision Letter · Decision Letter 0]

5 Dec 2025

Dear Dr. Al-Qahtani,

Thank you for submitting your manuscript to PLOS ONE. After careful consideration, we feel that it has merit but does not fully meet PLOS ONE’s publication criteria as it currently stands. Therefore, we invite you to submit a revised version of the manuscript that addresses the points raised during the review process.

We look forward to receiving your revised manuscript.

Kind regards,

Chandrabose Selvaraj, Ph.D.

Academic Editor

PLOS One

Journal Requirements:

“This research work was funded by the Institutional Fund Projects under grant number IFPNC: 009-117-2020. The authors gratefully acknowledge technical and financial support from the Ministry of Education and King Abdulaziz University, Jeddah, Saudi Arabia. The funders had no role in study design, data collection and analysis, decision to publish, or preparation of the manuscript.”

4. We note that your Data Availability Statement is currently as follows: “The Data used in this study are public, and the analysed data are in the manuscript as figures, tables and supplementary files.”

Additional Editor Comments:

**One of the reviewers has suggested citations that may not be directly relevant to the manuscript. The authors are encouraged to decline such recommendations, ensuring that all referenced works remain closely aligned with your research contributions. Declining these suggested citations will not affect the editorial decision on the manuscript.**

Reviewers' comments:

Reviewer's Responses to Questions

**Comments to the Author**

1. Is the manuscript technically sound, and do the data support the conclusions?

Reviewer #1: Yes

Reviewer #2: Yes

2. Has the statistical analysis been performed appropriately and rigorously?

Reviewer #1: N/A

Reviewer #2: Yes

3. Have the authors made all data underlying the findings in their manuscript fully available?

Reviewer #1: Yes

Reviewer #2: Yes

4. Is the manuscript presented in an intelligible fashion and written in standard English?

Reviewer #1: Yes

Reviewer #2: Yes

Reviewer #1: Dear Authors,

I read your manuscript, and it is wonderful. The work is very interesting, and at the same time, the way you presented figures, tables, and formatted the manuscript made it very easy to read and follow. However, there are still some changes or improvements that you need to make before it is accepted.

1. Mention the number of compounds taken for the docking in the Abstract, and do not make keywords go in italics.

2. Provide the list of abbreviations at the end of the manuscript in a single subheading.

3. Add a dedicated heading for the limitations and future directions of the work.

4. The conclusion has too much data to understand. Can you make the line 1096-1100 a bit smaller and clearer?

5. Funding and Acknowledgements should be in different points.

6. The supplementary file has too much data to report, but it is not cited in the manuscript. Authors are also recommended to cite the sheets.

7. RMSD, RMSF, and other abbreviations are explained multiple times. Keep the full form only at the first occurrence, then use the shorter form thereafter.

In my opinion, after addressing the above points, the manuscript is ready for acceptance.

Reviewer #2: Compare the HOMO-LUMO gap energy with biologically active molecules in the literature (10.1134/S0036024425701249, 10.1080/00387010.2024.2432331, 10.55730/1300-0527.3702).

Support the active site found by molecular docking and molecular dynamics calculations with experiments or literature.

**Do you want your identity to be public for this peer review?** For information about this choice, including consent withdrawal, please see our Privacy Policy

Reviewer #1: No

Reviewer #2: **Yes:** Sefa Celik

---

## [Decision Letter · Decision Letter 1]

7 Jan 2026

Precision Screening Identifies Mitoxantrone as a Multitarget Inhibitor in Ageing-Associated Cancers with Extensive Computational Validation and Doxorubicin Comparison

PONE-D-25-58708R1

Dear Dr. Al-Qahtani,

We’re pleased to inform you that your manuscript has been judged scientifically suitable for publication and will be formally accepted for publication once it meets all outstanding technical requirements.

Kind regards,

Chandrabose Selvaraj, Ph.D.

Academic Editor

PLOS One

Additional Editor Comments (optional):

Reviewers' comments:

Reviewer's Responses to Questions

**Comments to the Author**

Reviewer #1: All comments have been addressed

Reviewer #2: All comments have been addressed

2. Is the manuscript technically sound, and do the data support the conclusions?

Reviewer #1: Yes

Reviewer #2: Yes

3. Has the statistical analysis been performed appropriately and rigorously?

Reviewer #1: N/A

Reviewer #2: Yes

4. Have the authors made all data underlying the findings in their manuscript fully available?

Reviewer #1: Yes

Reviewer #2: Yes

5. Is the manuscript presented in an intelligible fashion and written in standard English?

Reviewer #1: Yes

Reviewer #2: Yes

Reviewer #1: The authors has resolved all of my comments by citing the supp sheets, providing list of abbreviations, adding a completed dedicated section for the Limitations and future work and many others.

Therefore I mark this paper to be accepted by PlosOne. Congratulations.

Reviewer #2: The authors have implemented the requested revisions.

**Do you want your identity to be public for this peer review?** For information about this choice, including consent withdrawal, please see our Privacy Policy

Reviewer #1: No

Reviewer #2: No

---

## [Editor Report · Acceptance letter]

PONE-D-25-58708R1

PLOS One

Dear Dr. Al-Qahtani,

I'm pleased to inform you that your manuscript has been deemed suitable for publication in PLOS One. Congratulations! Your manuscript is now being handed over to our production team.

Kind regards,

on behalf of

Dr. Chandrabose Selvaraj

Academic Editor

PLOS One